# The role of *Xist*-mediated Polycomb recruitment in the initiation of X-chromosome inactivation

Aurélie Bousard[1,†], Ana Cláudia Raposo[2,†], Jan Jakub Żylicz[1,3,†], Christel Picard[1], Vanessa Borges Pires[2,4], Yanyan Qi[5], Cláudia Gil[2], Laurène Syx[1], Howard Y Chang[5,6], Edith Heard[1,*] iD & Simão Teixeira da Rocha[1,2,**] iD

## Abstract

*Xist* RNA has been established as the master regulator of X-chromosome inactivation (XCI) in female eutherian mammals, but its mechanism of action remains unclear. By creating novel *Xist*-inducible mutants at the endogenous locus in male mouse embryonic stem (ES) cells, we dissect the role of the conserved A-B-C-F repeats in the initiation of XCI. We find that transcriptional silencing can be largely uncoupled from Polycomb repressive complex 1 and complex 2 (PRC1/2) recruitment, which requires B and C repeats. *Xist* ΔB+C RNA specifically loses interaction with PCGF3/5 subunits of PRC1, while binding of other *Xist* partners is largely unaffected. However, a slight relaxation of transcriptional silencing in *Xist* ΔB+C indicates a role for PRC1/2 proteins in early stabilization of gene repression. Distinct modules within the *Xist* RNA are therefore involved in the convergence of independent chromatin modification and gene repression pathways. In this context, Polycomb recruitment seems to be of moderate relevance in the initiation of silencing.

**Keywords** chromatin; PRC1; PRC2; X-chromosome inactivation; *Xist*
**Subject Categories** Chromatin, Transcription & Genomics; RNA Biology

## Introduction

Long non-coding RNAs (lncRNAs) are a class of non-protein coding RNAs of > 200 nucleotides that are frequently capped, spliced, and polyadenylated. Some are located in the nucleus and have been implicated in transcriptional regulation and recruitment of chromatin modifiers, using still poorly defined molecular mechanisms (reviewed in refs. [1,2]). *Xist* (*X-inactive-specific transcript*) lncRNA represents the most studied paradigm of a nuclear RNA with documented roles in transcription regulation and recruitment of chromatin modifiers in female eutherian mammals (reviewed in ref. [3]). *Xist* lncRNA is ultimately expressed from only one of the two X chromosomes, "coating" in *cis* its chromosome territory and triggering a cascade of events that result in chromosome-wide gene silencing and formation of facultative heterochromatin (reviewed in ref. [3]). How *Xist* coordinates these two processes, and their causal relationship, is still unclear. In this context, the Polycomb group (PcG) proteins are of particular interest (reviewed in refs. [4,5]).

Recruitment of PcG proteins following *Xist* RNA coating is an early event during XCI. Both PRC1 and PRC2 are recruited to lay down H2AK119ub and H3K27me3 on the future inactive X chromosome (Xi), respectively [6–8]. Recently, our refined temporal analysis showed that PRC1-associated H2AK119ub mark precedes the accumulation of the PRC2-associated H3K27me3 mark on the inactivating X chromosome [9]. Both canonical PRC1, with a CBX subunit, and non-canonical versions, with RYBP/YAF2 subunits, are known to be recruited to the Xi [10–12]. Recently, several studies showed that the non-canonical PCGF3-PRC1 and PCGF5-PRC1 complexes associate with *Xist* RNA via hnRNPK RNA binding protein (RBP) [10,13–15]. PCGF3 and PCGF5 are paralogs, and both PCGF3-PRC1 and PCGF5-PRC1 complexes are very similar in function [10,16] and composition: Besides PCGF3 or PCGF5, they contain the catalytic RING subunits RING1A/RING1B, RYBP/YAF2, FBRS, and the accessory proteins CK2 and AUTS2 [17,18]. For this reason, they will be defined as PCGF3/5-PRC1 throughout this article. PCGF3/5-PRC1 appears to mediate early H2AK119ub deposition [10] which seems to be sensed by the JARID2, a PRC2 co-factor, that helps to bring PRC2 to the Xi [19,20]. However, this interdependent model for PcG recruitment might not fully explain how PRC1 and PRC2 are recruited to the X chromosome. Indeed, independent pathways for PRC1 and PRC2 recruitment seem to also play a smaller

1  Mammalian Developmental Epigenetics Group, Institut Curie, CNRS UMR3215, INSERM U934, PSL University, Paris, France
2  Instituto de Medicina Molecular João Lobo Antunes, Faculdade de Medicina, Universidade de Lisboa, Lisboa, Portugal
3  Department of Physiology, Development and Neuroscience, University of Cambridge, Cambridge, UK
4  Instituto de Ciências Biomédicas Abel Salazar, Universidade do Porto, Porto, Portugal
5  Center for Dynamic Personal Regulomes, Stanford University, Stanford, CA, USA
6  Howard Hughes Medical Institute, Stanford University, Stanford, CA, USA
   *Corresponding author. Tel: +49 62213870; E-mail: edith.heard@curie.fr
   **Corresponding author. Tel: +351 217999411; E-mail: simaoteixeiradarocha@medicina.ulisboa.pt
   †These authors contributed equally to this work

role [14], highlighting the fact that mechanisms of *Xist*-mediated PcG recruitment are far from being fully understood.

The role of PRC2 in XCI was first uncovered through the analysis of a mouse knock-out for *Eed*, a PRC2 component, in which loss in maintenance of XCI was seen in the extra-embryonic tissues of female embryos [21,22]. In contrast, the role of PRC1 or PRC2 during XCI in embryonic lineages or in differentiating ES cells remains inconclusive with slightly inconsistent results [8,10,11,23]. These discrepancies arise in part from the models used to address this question. *In vivo* analysis of XCI has been confounded by severe developmental abnormalities and early lethality upon disruption of PRC2 function [8,23]. Similarly, *in vitro* analyses usually involved *Xist* transgenes on autosomes which may not fully recapitulate the chromatin requirements of the X chromosome during XCI [10,11,24]. Moreover, deletion of PcG proteins delays ES cell differentiation [20,25,26], a necessary precondition for *Xist* expression and XCI. It is in this context that we set out to address PcG function in *Xist*-dependent transcriptional silencing during early ES cell differentiation.

*Xist* is an unusually long RNA (15,000–17,000 nt) with low overall sequence conservation, except for a series of unique tandem repeats, named A-to-F (Fig 1A) [27–29]. The most conserved and best studied is the A repeat, which is essential for *Xist*-mediated gene silencing [30]. The A repeat interacts specifically with proteins such as SPEN and RBM15 both believed to be involved in its gene silencing role [13,31–35]. Other *Xist* RNA repeat regions have been implicated in the recruitment of factors involved in *cis*-localization (*e.g.*, recruitment of CIZ1 matrix attachment protein by the E repeat) [36,37] or Polycomb chromatin modifications [14,15,20]. We previously showed that a region spanning F, B, and C repeats is critical for PRC2 recruitment to the Xi [20]. More recently, using an autosomal *Xist* transgene, it was reported that a 600 bp *Xist* region containing the B repeat was necessary for PRC2 and PRC1 recruitment through direct binding of hnRNPK RBP [15]. The B repeat was also found to be important for *Xist* spreading and continuous recruitment of PRC1/2 during the maintenance stage of XCI [14].

The exact contributions of *Xist*'s repeat regions to the initiation of XCI have remained unclear due to three main issues: (i) Some of the deletions at the endogenous *Xist* locus can impair expression of the mutant allele and/or lead to skewed XCI toward the wild-type allele [38–40]; (ii) deletions of the repeat elements can result in delocalization of *Xist* from the Xi territory, which indirectly affects gene silencing and chromatin changes [14,36,37,41]; and (iii) deletions performed in the context of autosomal cDNA-inducible systems are difficult to interpret due to the reduced efficiency of *Xist*-mediated silencing of autosomal genes [15,24,42].

In this study, we generate and analyze a series of *Xist* mutants created at the endogenous *Xist* gene under an inducible promoter. In particular, we explored the endogenous *Xist* RNA's sequence requirements for recruitment of PRC1/PRC2 and re-assessed the relationship between the initiation of X-linked transcriptional silencing and PcG recruitment. Our results reveal that removal of both *Xist* B and C repeats, and not just the B repeat as previously proposed in the context of autosomal-inducible transgenes or transformed mouse embryonic fibroblasts [14,15], is necessary to fully abolish PRC1/PRC2 recruitment in the context of the X chromosome in differentiating *Xist*-induced ESCs. Moreover, we provide evidence

that X-linked transcriptional silencing can occur in a PcG-defective-inducible *Xist* mutant, albeit slightly less efficiently.

# Results

### Generation of *Xist* RNA mutants for F, B, and C repeats

To dissect the role of different functional RNA domains of *Xist*, particularly the RNA sequences enabling recruitment of PRC1 and PRC2 complexes to the X chromosome, we created a series of new inducible *Xist* mutants. For this, we used a previously described system whereby *Xist* is driven by a tetracycline-inducible promoter (*Xist*-TetOP) that can be activated by doxycycline (DOX) at its endogenous locus in J1 XY embryonic stem (ES) cells (Fig 1A). This system recapitulates hallmarks of XCI, namely chromosome-wide *Xist* coating, X-linked gene silencing, and heterochromatin formation [30], and has been extensively used in the literature [9,30,43–46]. We created 6 new mutants within *Xist* exon 1: ΔF+B+C, ΔF+B, ΔB+C, ΔB+1/2C, ΔB, and ΔC by CRISPR/Cas9 genome editing using flanking pairs of guide RNAs (gRNAs) (Fig 1A; Materials and Methods). At least two clones per type of mutation were created (Table EV1). The previously generated *Xist* ΔA mutant, which is silencing-defective [30], but competent for PRC2 recruitment [20], was also used in this study for comparison (Fig 1A).

The newly generated *Xist* mutants were validated at the DNA and RNA level by PCR and RT–PCR, and the exact deleted regions were mapped by Sanger sequencing (Fig EV1A and B). RNA fluorescent *in situ* hybridization (FISH) analysis showed that all *Xist* mutants are able to form a *Xist* domain upon DOX induction, but not in non-induced (noDOX) conditions (Fig EV1C). The proportion of cells with a *Xist*-coated X chromosome varied somewhat between the mutant clones [*e.g.*, *Xist* FL: 45 ± 6%; *Xist* ΔA: 53 ± 9%; and *Xist* ΔB+C: 60 ± 8% at day 4 of differentiation in DOX conditions; Fig EV1C]. This is not surprising since variation in the proportion of *Xist*-coated X chromosomes has been previously appreciated for the *Xist* FL and *Xist* ΔA lines [9,20,44,45]. The two clones of each mutant type did not always have the same percentage of cells with *Xist* domains (Table EV1), suggesting that the differences between the lines are unlikely to be explained by *Xist* mutant type, but rather by the variable ability of cell lines to respond to DOX. Next, we employed this new series of *Xist*-TetOP mutants to assess their XCI and chromatin-associated phenotypes.

### PcG complexes are recruited to the X chromosome thanks to B and C repeats of *Xist*

Previously, our results and those of others have shown that PRC1/PRC2 recruitment is impaired in the *Xist* ΔXN cDNA mutant that lacks 3.8 kb including the F, B, and C repeats in differentiating ESCs [10,20]. More recently, a 600-bp region including the B repeat as well as the first 3 of the 14 motifs of the C repeat was deleted and reported to abrogate Polycomb recruitment, although this was in the context of autosomal *Xist*-inducible cDNA transgenes [15]. What is more, the B repeat has also been implicated in continuous PcG recruitment in the maintenance stage of XCI based on results using transformed tetraploid mouse embryonic fibroblasts (MEFs) [14]. To assess this in the context of silencing initiation at the

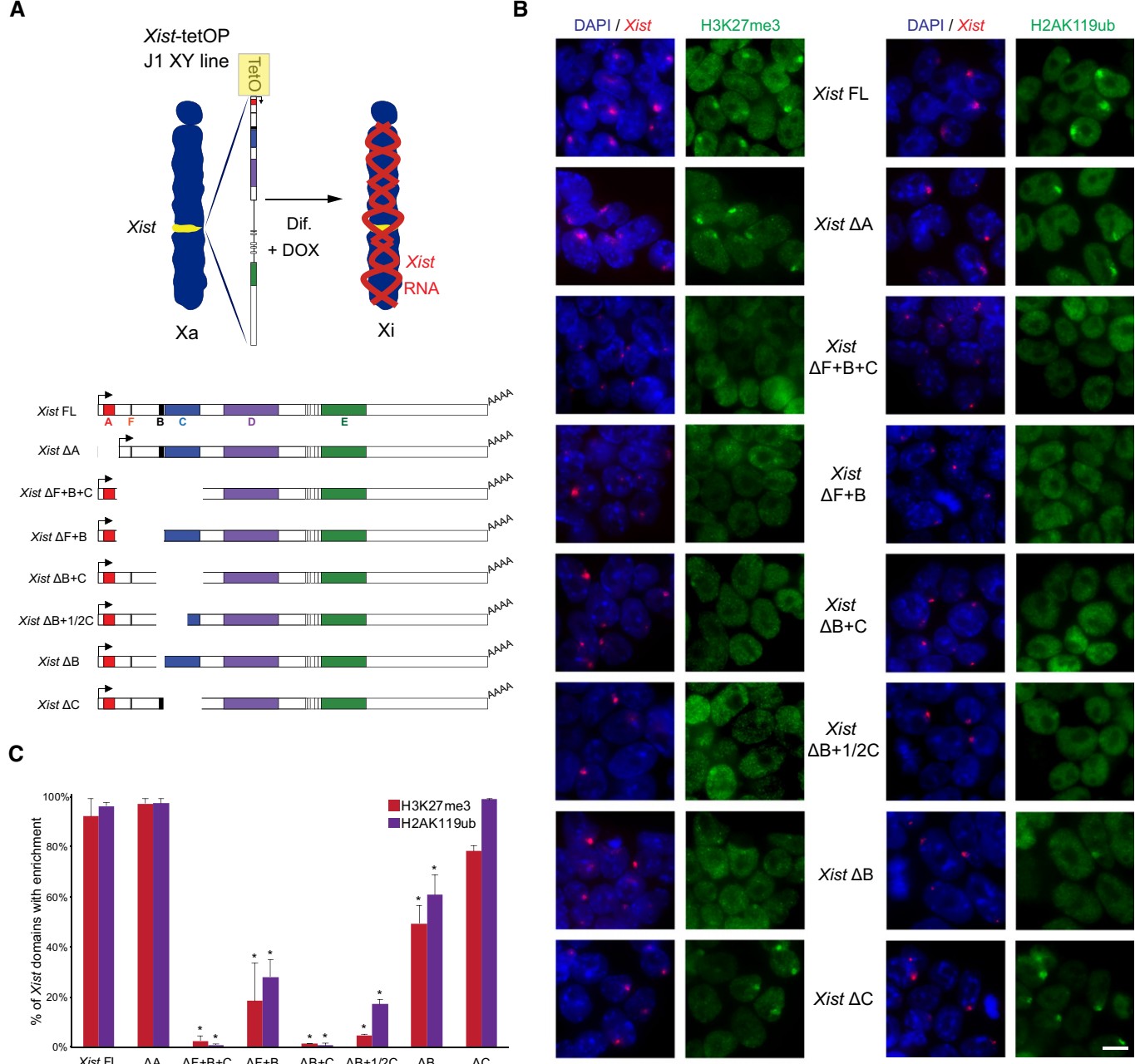

**Figure 1. Lack of H3K27me3 and H2AK119ub enrichment over the X chromosome in the absence of *Xist* repeats B and C.**

A  Schematic representation of the novel *Xist*-TetOP mutants generated by CRISPR/Cas9 genome editing in J1 XY ESCs; the different repeats are highlighted in color boxes; Dif.—differentiation; DOX—doxycycline.

B  Representative images of combined IF for H3K27me3 or H2AK119ub (green) with RNA FISH for *Xist* (red) in *Xist*-TetOP lines (for clone 1 of each mutant type) at day 2 of differentiation in DOX conditions; blue—DAPI staining; scale bar: 10 μm.

C  Graph represents the mean % + SEM of *Xist*-coated X chromosomes enriched for H3K27me3 or H2AK119ub in the different *Xist*-TetOP mutants (for clone 1 of each mutant type) from 2 to 4 independent experiments; a minimum of 50 *Xist*-coated X chromosomes were counted per experiment; only *P*-values corresponding to significant differences from unpaired Student's *t*-test comparing mutants to *Xist* FL are indicated as * (*P*-value < 0.05).

Source data are available online for this figure.

X chromosome, we evaluated whether the different *Xist*-TetOP mutants exhibited typical H3K27me3 and H2AK119ub foci over the *Xist*-coated X chromosome. For this, we performed combined immunofluorescence (IF)/*Xist* RNA FISH at day 2 of differentiation in the presence of DOX, a time point where PcG recruitment reaches its maximum [20]. Enrichment of H3K27me3 and H2AK119ub was seen at the *Xist*-coated X chromosome in most cells in the *Xist* FL and *Xist* ΔA cell lines (Fig 1B and C), consistent with previous

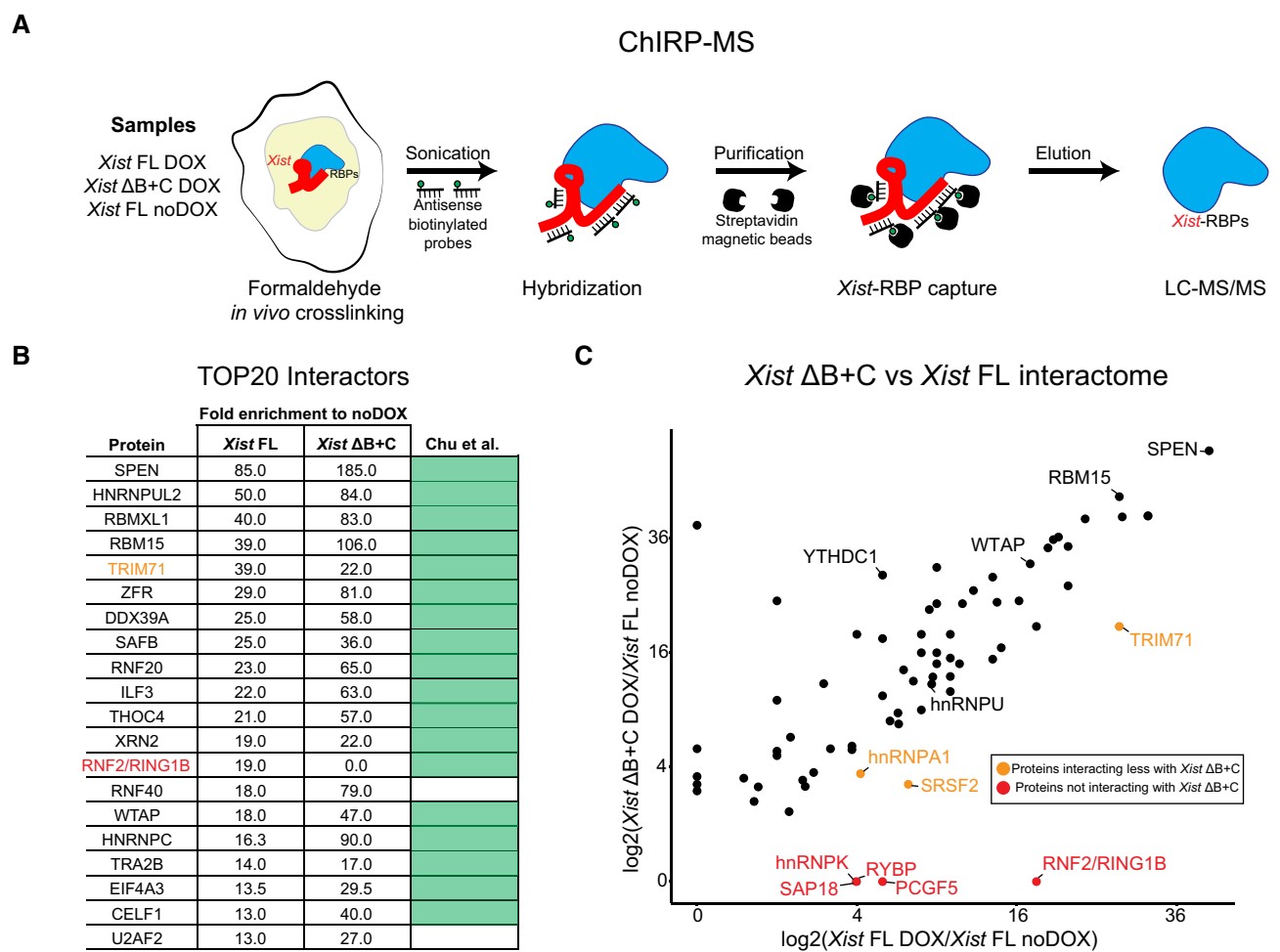

**Figure 2. Absence of PCGF3/5-PRC1 proteins from the *Xist* ΔB+C RNA protein interactome.**

A  Scheme of the ChIRP-MS workflow performed on *Xist* FL (DOX and noDOX conditions) and *Xist* ΔB+C (DOX) at day 3 of differentiation; RBP—RNA-binding protein.

B  Top 20 protein hits from the ChIRP-MS of *Xist* FL; the ranking was based on fold enrichment of *Xist* FL DOX versus *Xist* FL noDOX; weakly annotated protein isoforms with an Annotation score in UniProtKB < 3 (out of 5) were excluded; fold enrichment for *Xist* ΔB+C is also displayed for comparison; light green boxes correspond to proteins previously described by Chu et al [13] as *Xist* interactors; protein in red (RNF2/RING1B) represents a protein not found in the *Xist* ΔB+C interactome; protein in light brown (TRIM71) is less enriched in *Xist* ΔB+C than in *Xist* FL.

C  Scatter plot displaying the differences in peptide counts between *Xist* FL and *Xist* ΔB+C for the 74 out of 81 *Xist* interactors from Chu et al [13] with a minimum of fold change of 2.5 in *Xist* FL or *Xist* ΔB+C; shown is the log2 fold change in peptide counts of each mutant in DOX conditions compared with the *Xist* FL in noDOX conditions; proteins retrieved by both *Xist* FL and *Xist* ΔB+C ChIRPs with a proposed role in XCI such as SPEN, RBM15, WTAP, YTHDC1, and hnRNPU are indicated; light brown dots mark proteins more represented in *Xist* FL than in *Xist* ΔB+C ChIRPs, while red dots display proteins which are only retrieved by *Xist* FL ChIRP.

reports [10,20,32]. Interestingly, lack of the *Xist* C repeat alone did not significantly affect H3K27me3 or H2AK119ub enrichment. In contrast, all *Xist*-TetOP mutants for which B repeat was absent showed a statistically significant decrease in H3K27me3 and H2AK119ub over the *Xist*-coated X chromosome (Fig 1B and C). Nevertheless, a slight enrichment of these marks was still seen in around half of *Xist* domains in *Xist* ΔB, to a lesser degree in *Xist* ΔF+B and even less in *Xist* ΔB+1/2C, which lacks 62% of the C repeat. In the *Xist* ΔB+C and ΔF+B+C mutants, no H3K27me3 and H2AK119ub enrichment was observed (Fig 1B and C). These results were confirmed in the clone 2 for each mutant type (Table EV2). The defects in the enrichment of PcG-associated histone modifications were associated with reduced recruitment of PRC2 (EZH2) and its co-factor (JARID2) and of PRC1 (RING1B; Fig EV2A and B;

Table EV2). These defects were more pronounced, likely because histone marks are stably maintained while the PcG complexes are dynamically recruited. All in all, our results show that PRC1 and PRC2 require the same *Xist* RNA modules to enable recruitment to the X chromosome. This is consistent with the dependence of PRC2 recruitment on the non-canonical PRC1 [10,15,19,20]. Furthermore, we show that the deletion of both B and C repeats is needed to completely abrogate PcG recruitment in the context of inducible *Xist* expression from its endogenous locus. This is a significantly bigger region than that necessary to cause the same defect in the context of autosomal-inducible *Xist* transgene integrations (2.1 kb versus 0.6 kb) [15]. Thus, the severity of phenotypes seems to depend on the chromosomal context where *Xist*-dependent gene silencing is induced.

### Xist ΔB+C RNA does not interact with PCGF3/5-PRC1

To obtain mechanistic insight into why Xist ΔB+C RNA does not recruit PcG proteins globally, we first assess whether the stability of the mutant RNA was affected by evaluating Xist half-life upon RNA polymerase II transcriptional inhibition through actinomycin D treatment. No discernible differences were seen between Xist FL and Xist ΔB+C (Fig EV2C), suggesting comparable levels of RNA stability. Then, we analyzed the pattern of Xist ΔB+C coating by performing RNA FISH using customized Stellaris oligo-probes not overlapping the deleted region (see Materials and Methods). Both Xist cloud size and total intensity of the signal were not statistically different between Xist FL and Xist ΔB+C RNA-coated chromosomes (Fig EV2D and E). This suggests that the PcG recruitment phenotype is not due to incorrect Xist ΔB+C coating. Interestingly, we noted that the size and total signal intensity of clouds were higher when compared to the Xist cloud of the stable Xi in female primary mouse embryonic fibroblasts (MEFs) (Fig EV2D and E). This might indicate an increase in Xist molecules surrounding the Xi in our inducible system compared to normal Xist coating during XCI maintenance, which, nonetheless, does not differ between Xist FL and Xist ΔB+C RNAs.

Given the comparable stability and cloud signal among Xist FL and Xist ΔB+C RNAs, we then analyzed the protein interactome of the Xist ΔB+C RNA using ChIRP-MS (RNA-binding proteins by mass spectrometry). Previously, ChIRP-MS identified 81 Xist protein partners, three of which (SPEN, WTAP, and RNF20) bind to the A repeat [13]. We performed ChIRP-MS on both Xist FL and Xist ΔB+C cells in induced (DOX) conditions at day 3 of differentiation as previously performed for Xist FL and Xist ΔA differentiated ES cells [13]. As a negative control, Xist FL ES cells were also differentiated in noDOX conditions (Fig 2A). We confirmed that Xist RNA was retrieved after ChIRP procedure in DOX, but not in noDOX conditions (Fig EV3A). The Xist RNA levels recovered from Xist ΔB+C were higher than from Xist FL-induced cells (Fig EV3B), due to a higher proportion of cells inducing Xist (50.9% versus 24.0%; Fig EV3B) but not because of increased RNA induction levels in individual cells (Fig EV2D and E). Proteins retrieved by Xist ChIRP were separated by electrophoresis (Fig EV3C) and sent for identification by liquid chromatography–tandem mass spectrometry (LC-MS/MS).

Previously described Xist protein interactors [13] were found among the top hits in Xist FL RNA (and also in Xist ΔB+C RNA) confirming the success of the ChIRP-MS experiment (Dataset EV1). Indeed, considering the 20 top hits for Xist FL RNA after filtering out weakly annotated protein isoforms, 18 of them are in the Chu et al list [13]. Of these top 20 hits, all were shared with Xist ΔB+C RNA, with the notable exception of the PRC1 component RNF2/RING1B (Fig 2B; Dataset EV1). Overall, higher fold enrichment for the remaining factors in Xist ΔB+C compared to Xist FL is consistent with the increased yield of Xist RNA and proteins retrieved from mutant cells (Figs 2B and EV3A–C; Dataset EV1).

By focusing on the 81 hits previously identified by Chu et al [13], we compared the protein interactomes of Xist FL and Xist ΔB+C RNAs (Fig 2C). Consistently with our previous analysis, we detected the majority of published hits to interact with both Xist FL and Xist ΔB+C (74 out of 81 with a minimum of 2.5

DOX/noDOX fold change in one of the samples) (Dataset EV1). SPEN and many other proteins with a proposed role in long-range gene silencing were present in the Xist ΔB+C interactome, such as members of the m6A RNA methyltransferase machinery (RBM15, WTAP, and YTHDC1) and proteins involved in Xist spreading such as the hnRNPU matrix attachment protein (Fig 2C). In contrast, five proteins were absent from the Xist ΔB+C interactome (Fig 2C), including the three members of non-canonical PRC1 present in Chu et al's list—RNF2/RING1B, RYBP, and PCGF5. We also found that PCGF3, which was not in the original Chu et al's list, was present in the Xist FL interactome, but lacking from the Xist ΔB+C interactome in our ChIRP-MS experiment (Dataset EV1). Furthermore, hnRNPK RBP previously linked to Xist-induced PCGF3/5-PRC1 recruitment, was also not detected in Xist ΔB+C ChIRP-MS (with the exception of two poorly annotated isoforms) (Dataset EV1), corroborating findings from others using in vitro pull-down experiments [14,15]. The histone deacetylase complex subunit SAP18 was also lacking from the Xist ΔB+C interactome, while three other proteins were found to bind more weakly to Xist ΔB+C than to Xist FL RNA: TRIM71 (an E3 ubiquitin-protein ligase); SRSF2 (serine and arginine-rich splicing factor 2); and hnRNPA1 (heterogeneous nuclear ribonucleoprotein A1; Fig 2C; Dataset EV1).

In conclusion, Xist ΔB+C and Xist FL RNAs share most of their protein interactome with few exceptions such as proteins of the PCGF3/5-PRC1 complex. Combined with previous results on Xist ΔA [13], these data illustrate the modular organization of Xist lncRNA, with RNA motifs interacting independently with different proteins and possibly performing distinct functions. The absence of PCGF3/5-PRC1 from the Xist ΔB+C interactome explains the global lack of H2AK119ub and concomitant loss of H3K27me3 enrichment over the X chromosome (Fig 1B and C), consistent with the hierarchical model proposed for PRC1/PRC2 recruitment [9,10].

### General depletion of PcG marks across the Xist ΔB+C-coated X chromosome with minor accumulation over genes

To assess the lack of H3K27me3 and H2AK119ub accumulation at the chromosome-wide level in the Xist ΔB+C mutant differentiating ES cells, we performed native ChIP-seq (nChIP-seq) for these marks. Both marks were assessed after 2 days of differentiation in DOX and noDOX conditions in biological duplicates for Xist ΔB+C mutant cells and compared to the results previously obtained for Xist FL cells [9]. At autosomal sites, similar patterns of enrichment for the PcG marks were observed for all the samples in both DOX and noDOX conditions (e.g., HoxC cluster in Fig EV4A). At the level of the X chromosome, we observed a general loss of H3K27me3 and H2AK119ub accumulation in the Xist ΔB+C mutant in clear contrast to Xist FL (Figs 3A and EV4B). However, we noted some residual accumulation for both marks in Xist ΔB+C at gene-dense regions which are normally active in noDOX conditions (Fig 3A). We therefore evaluated enrichment at specific types of genomic regions: promoters and gene bodies which were initially active in noDOX conditions (herein called as initially active promoters and gene bodies, respectively; see Materials and Methods for definition) and intergenic regions. Consistent with the chromosome-wide analysis, at intergenic windows, we observed a striking lack of H3K27me3

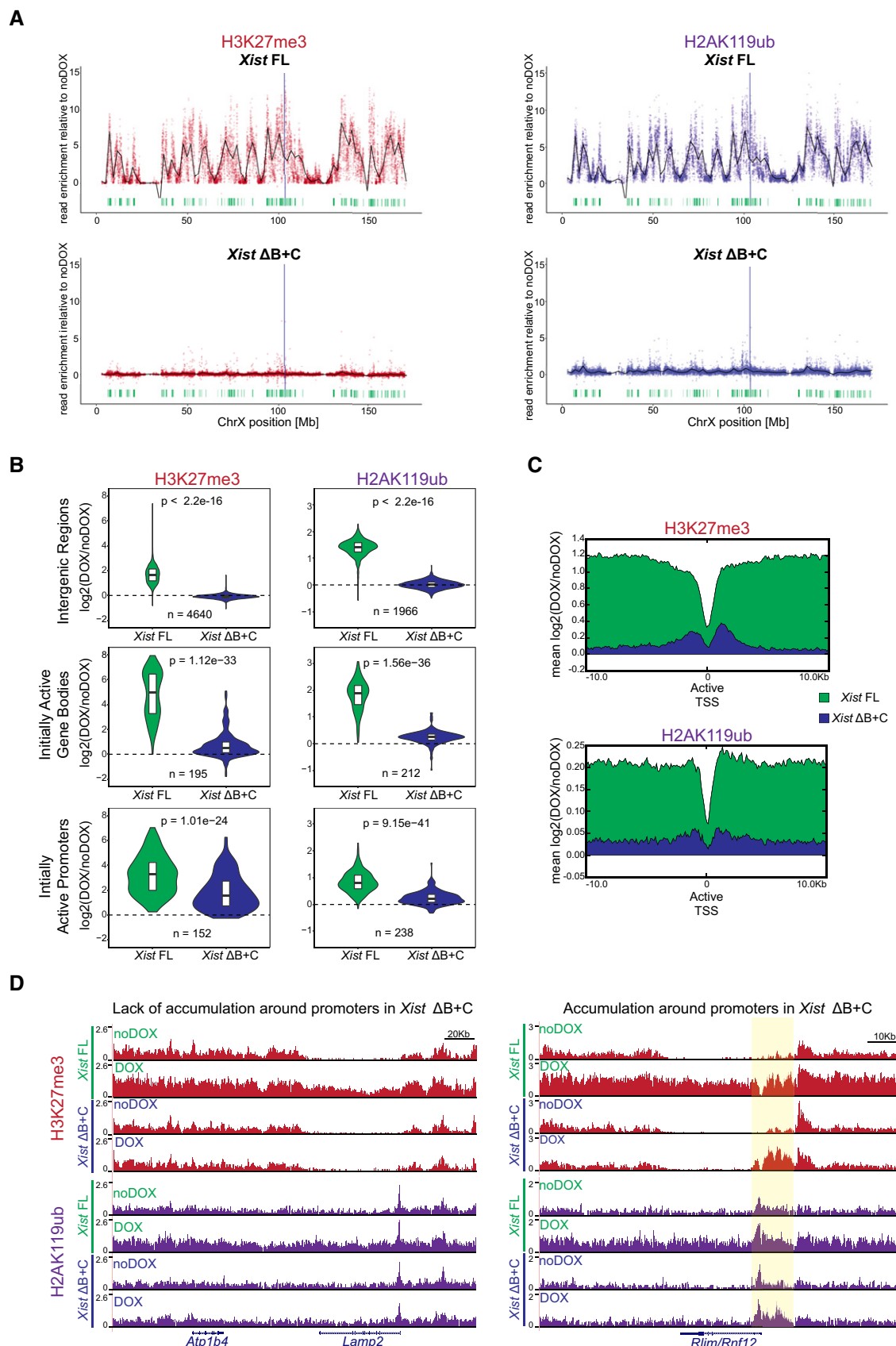

**Figure 3.**

◄

**Figure 3.   nChIP-seq reveals chromosome-wide absence of H3K27me3 and H2AK119ub enrichment, but residual enrichment at initially active genes in the *Xist* ΔB+C.**

A    Plots showing H3K27me3 and H2AK119ub accumulation over the X chromosome in *Xist* FL and *Xist* ΔB+C cell lines upon DOX induction at day 2 of differentiation; each dot represents a single 10-kb window and its enrichment relative to noDOX condition; black line is a loess regression on all windows; *Xist* locus is represented by a blue long line, and active genes by green lines.

B    Violin plots quantifying H3K27me3 and H2AK119ub enrichment over intergenic regions, initially active promoters, and initially active gene bodies in the X chromosome in *Xist* FL and *Xist* ΔB+C cell lines at D2 upon DOX induction; shown is the $\log_2$ fold change in DOX versus noDOX conditions; violin plots represent the distribution of the values, the horizontal band is the median, and the lower and upper hinges correspond to the 25th and 75th percentiles; *n* = indicates the number of regions/genes analyzed; *P*-values were calculated using a paired Wilcoxon test, comparing *Xist* FL and *Xist* ΔB+C cell lines.

C    Average plots showing the mean enrichment of H3K27me3 (top) and H2AK119ub (bottom) over all X-linked initially active transcriptional start sites (TSS); shown is the mean of normalized log2 enrichment of DOX versus noDOX in both *Xist* FL and *Xist* ΔB+C cell lines.

D    Genome browser plots showing H3K27me3 (top) and H2AK119ub (bottom) enrichments in a region encompassing the inactive *Atp1b4* and the initially active *Lamp2* genes within the XqA3.3 region and the initially active *Rlim/Rnf12* gene at the XqD region; region around the promoter of *Rlim/Rnf12* is highlighted in yellow.

Source data are available online for this figure.

and H2AK119ub enrichment upon induction of *Xist* ΔB+C when compared to *Xist* FL RNA (Fig 3B). In contrast, slight enrichment of both PcG-associated marks was detected upon *Xist* ΔB+C induction at initially active promoters and gene bodies, in particular for H3K27me3 over initially active promoters (Figs 3B and EV4C). This enrichment of both marks was significantly lower than that observed in *Xist* FL expressing cells (Fig 3B), as can be visualized using average plots around transcriptional start sites (TSS) of initially active genes (Fig 3C). We also normalized our data for the percentage of cells presenting *Xist*-coated chromosomes, based on RNA FISH analysis (Fig EV4D) and obtained similar results (Fig EV4E). Examples of typical nChIP-seq profiles are depicted in Fig 3D showing a gene with lack of accumulation for PcG marks (*Lamp2*), and a second gene (*Rlim/Rnf12*) with clear H3K27me3/H2AK119ub enrichment around the promoter in the induced *Xist* ΔB+C mutant cells. In conclusion, we observed no enrichment of H3K27me3 and H2AK119ub at intergenic regions upon expression of *Xist* ΔB+C RNA, but a mild accumulation is seen over some initially active promoters and to a lesser extent at initially active gene bodies. As genes represent only a small fraction of the X chromosome, this is probably why we could not detect their enrichment in the IF/RNA FISH experiments. The reasons behind this minor enrichment of H3K27me3 and H2AK119ub at some X-linked genes in the *Xist* ΔB+C mutant that cannot bind PCGF3/5-PRC1 proteins are unclear, but could be due to the transcriptional silencing of these genes.

**Xist ΔB+C RNA is able to initiate long-range transcription silencing along the X chromosome**

To assess the degree to which transcriptional silencing could be induced in the *Xist* ΔB+C expressing cells, we evaluate expression from X-linked genes. We initially performed nascent transcript RNA FISH combined with *Xist* RNA FISH for X-linked genes (*Pgk1* and *Lamp2* at D2; *Pgk1* and *Rlim*/*Rnf12* at D4) in different *Xist* mutant lines (Fig 4A and B; Table EV3). As expected, *Xist* ΔA RNA was entirely defective in silencing for the assessed X-linked genes. In striking contrast, all the other *Xist* mutants (ΔF+B+C, ΔF+B, ΔB+C, ΔB+1/2C, ΔB, and ΔC) were able to silence these genes at levels approximately similar to *Xist* FL RNA (Fig 4A and B; Table EV3). Similar results were obtained for the second clone of each mutant (Table EV3). Corroborating the nascent transcript RNA FISH data, we also noted significant

reduction in cell survival upon prolonged DOX induction (≥5 days) for *Xist* FL and all the mutants with the exception of *Xist* ΔA RNA (Table EV4). This is consistent with efficient XCI in XY ESCs, resulting in functional nullisomy for the X chromosome, and thus cell death. Interestingly, a mild relaxation of silencing could be seen for some genes, as, for example, *Lamp2* at D2 in PcG-defective *Xist* ΔF+B+C and *Xist* ΔB+C, but not in *Xist* FL, *Xist* ΔB, and *Xist* ΔC (Fig 4A).

To assess the full extent of transcriptional silencing of X-linked genes in the absence of *Xist*-mediated PcG recruitment, we examined RNA-seq on biological duplicates of *Xist* FL, *Xist* ΔA (silencing-defective), and *Xist* ΔB+C (PcG-defective) in DOX and noDOX conditions at day 2 of differentiation. We confirmed robust *Xist* upregulation upon DOX treatment and found no reads mapping to the deleted regions in both mutant lines (Fig EV5A). First, we evaluated whether the percentage of total X-chromosome-specific RNA-seq reads changed before and upon induction. While no changes were observed for the silencing-defective *Xist* ΔA cell line, the percentage of X-chromosome-specific reads decreased in both *Xist* FL and *Xist* ΔB+C cell lines upon DOX induction (Fig EV5B). Clustering analysis based on X-linked gene expression shows that DOX-induced samples of *Xist* FL and *Xist* ΔB+C segregate from the noDOX samples and DOX-induced *Xist* ΔA samples (Fig 4C). Furthermore, both *Xist* FL and *Xist* ΔB+C RNAs, but not *Xist* ΔA, were able to silence most genes throughout the X chromosome (Fig 4D). This is consistent with our previous nascent transcript RNA FISH analysis (Fig 4A and B). When we compared the average degree of silencing, we observed a slight relaxation of X-linked gene silencing in the *Xist* ΔB+C mutant when compared to *Xist* FL expressing cells (Fig 4E). This becomes more evident when data are adjusted for the percentage of cells presenting *Xist*-coated chromosomes as judged by RNA FISH (Fig EV5C and D). Nonetheless, this effect on gene silencing was significantly milder than that observed for *Xist* ΔA mutant expressing cells (Figs 4E and EV5D). The slight silencing defect in *Xist* ΔB+C expressing cells appears to be a chromosome-wide effect, since we could not pinpoint specific genes driving the differences in silencing efficiency between *Xist* ΔB+C and *Xist* FL (Fig EV5E). All in all, these results show that *Xist* ΔB+C RNA is able to silence X-linked genes, but the degree of overall silencing is less effectively initiated and/or maintained.

Finally, we wished to explore the relationship between PcG recruitment at promoters and initiation of X-linked gene

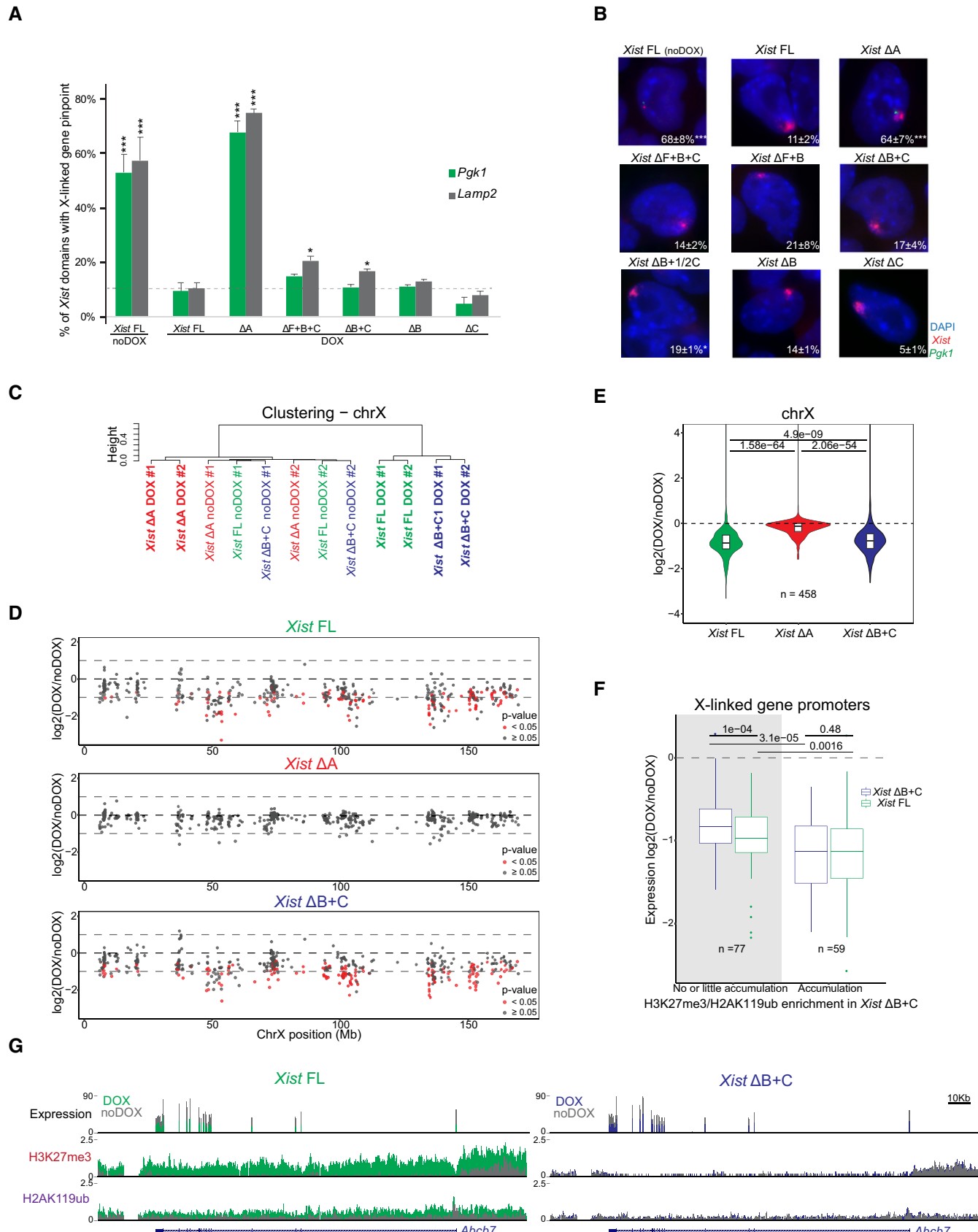

**Figure 4.**

**Figure 4. *Xist* ΔB+C is able to initiate X-chromosome-wide transcriptional silencing with no or residual Polycomb recruitment.**

A   Graph represents the mean % + SEM of *Xist*-coated chromosomes presenting an active *Pgk1* or *Lamp2* gene as determined by RNA FISH (as represented in B) at day 2 of differentiation in the presence of DOX (*Xist* FL was also used in noDOX conditions) in the different *Xist*-TetOP mutants; each bar represents the mean from 2 to 4 independent experiments; a minimum of 59 *Xist*-coated chromosomes were counted per experiment; for *Xist* FL noDOX, a minimum of 100 cells (which do not have *Xist*-coated chromosome) were counted; only *P*-values corresponding to significant differences comparing mutants (or *Xist* FL noDOX) to *Xist* FL DOX are indicated as * ($P < 0.05$) or *** ($P < 0.01$), unpaired Student's *t*-test; dashed line marks the mean percentage of silencing for the *Lamp2* gene in *Xist* FL DOX.

B   Representative RNA FISH images for *Xist* (red) and nascent transcript of *Pgk1* (green) in *Xist*-TetOP lines at day 4 of differentiation in the presence of DOX (*Xist* FL is also shown in noDOX conditions); DNA stained in blue by DAPI; numbers represent % of *Xist*-coated X chromosomes ± SEM with active *Pgk1* gene (except for *Xist* FL noDOX, where numbers represent % of cells with *Pgk1* active gene); the values represent 2–4 independent experiments, where a minimum of 50 *Xist*-coated chromosomes were counted per experiment; significant differences compared with *Xist* FL (DOX) are indicated as * ($P < 0.05$) or *** ($P < 0.01$), unpaired Student's *t*-test.

C   Clustering analysis of the normalized RNA-seq counts on the X chromosome (chrX) for all the duplicates of *Xist* FL, *Xist* ΔA, and *Xist* ΔB+C in DOX and noDOX conditions.

D   Plots display the $\log_2$ (fold change) in the expression of X-linked genes along the chrX comparing DOX versus noDOX samples for *Xist* FL, *Xist* ΔA, and *Xist* ΔB+C at day 2 of differentiation; red dots correspond to genes which are differently expressed in DOX versus noDOX ($P < 0.05$, Limma *t*-test), and while gray dots represent genes which are not differentially expressed between the two conditions ($P \geq 0.05$).

E   Violin plots displaying the average $\log_2$ (fold change) in gene expression between DOX and noDOX conditions on the chrX in *Xist* FL, *Xist* ΔA, and *Xist* ΔB+C at day 2 of differentiation; violin plots represent the distribution of the values, the horizontal band is the median, and the lower and upper hinges correspond to the 25th and 75th percentiles; *P*-values were calculated using a paired Wilcoxon test; $n$ = indicates the number of genes analyzed.

F   Box plots displaying the $\log_2$ (DOX/noDOX) fold change in expression of X-linked genes in *Xist* FL and *Xist* ΔB+C categorized according to the enrichment of H3K27me3 and H2AK119ub marks at promoters in *Xist* ΔB+C upon DOX induction (with no or little accumulation versus accumulation); the horizontal band of the box plot is the median of the values, the lower and upper hinges correspond to the 25th and 75th percentiles, the upper whisker extends from the hinge to the largest value not further than 1.5 interquartile range from the hinge, and the lower whisker extends from the hinge to the smallest value at most 1.5 interquartile range of the hinge; *P*-values between samples were calculated using a paired Wilcoxon test; $n$ = indicates the number of genes analyzed.

G   Genome browser plots showing RNA-seq reads, H3K27me3, and H2AK119ub nChIP reads around the *Abcb7* gene for *Xist* FL (left) and *Xist* ΔB+C (right) at day 2 of differentiation in both DOX and noDOX conditions.

Source data are available online for this figure.

silencing, given the slight enrichment of PcG marks over some X-linked genes in *Xist* ΔB+C-induced cells (Fig 3B and C). To address this, we categorized X-linked genes by their degree of silencing based on expression fold-change differences between DOX and noDOX conditions for both *Xist* FL and *Xist* ΔB+C. In both cases, accumulation of PcG marks at promoters correlated with the level of gene silencing (Fig EV5F). Within each of these categories of similarly silenced genes, H3K27me3 and H2AK119ub enrichment were significantly lower in *Xist* ΔB+C when compared to *Xist* FL (Fig EV5F). We next assessed whether genes that do not accumulate PcG marks upon *Xist* ΔB+C induction were silenced. We found 77 X-linked genes that accumulate little or no H3K27me3 and H2K119Aub marks at their promoters specifically in *Xist* ΔB+C-induced cells (Fig EV5G). These genes were nevertheless significantly silenced upon induction of the *Xist* ΔB+C RNA (Fig 4F) as exemplified by the *Abcb7* gene (Fig 4G). This suggests that PcG recruitment seems to be dispensable for initiating silencing of these genes. We noted, however, a slight silencing relaxation of these 77 genes when compared to *Xist* FL. Also, on average, these genes silenced less well than genes accumulating PcG marks in the mutant and *Xist* FL (Fig 4F). This implies that either PcG recruitment is needed to stabilize silencing initially imposed by other factors or that its mild, local enrichment of H3K27me3 and H2AK119ub is simply a consequence of X-linked gene silencing in *Xist* ΔB+C. The passive recruitment model is consistent with the fact that gene promoters accumulating PcG marks in *Xist* ΔB+C (and *Xist* FL) are enriched for CpG content (Fig EV5G and H). This feature is thought to promote PcG deposition at silenced promoters [47–49]. In conclusion, we believe our data point to a model, whereby *Xist*-mediated PcG accumulation via the B+C repeat region is not the initial driving force causing X-linked transcriptional silencing for most genes (Fig 5).

## Discussion

We found that chromosome-wide transcriptional silencing and PRC1/PRC2 recruitment rely primarily on the A and B+C *Xist* RNA repeats, respectively, on the X chromosome undergoing inactivation. Our analysis indicates that initiation of X-linked gene silencing can occur without *Xist*-induced chromosome-wide PRC1/PRC2 recruitment. However, PRC1/PRC2 and/or other proteins bound to the B+C region (e.g., SAP18; Fig 2C) seems to be necessary to stabilize the repressive state of some genes.

The inducible *Xist* mutants we have generated in this report represent a useful model for the study of individual *Xist* modules in the initiation of XCI, with *Xist* induction occurring at its endogenous location rather than at autosomal locations [15,24,42]. A clear advantage of these inducible systems for experimental purposes is that expression of *Xist* mutants can be induced and synchronized and does not depend on the intricate regulation of endogenous *Xist* expression which ultimately results in very few cells expressing the desired mutant [30,38–40]. A disadvantage is the increase in *Xist* RNA molecules surrounding the X chromosome (Fig EV2D and E) which could mask or attenuate few of the phenotypes. This has been controlled in this study by the use of the wild-type version (*Xist* FL) and a silencing-defective mutant, *Xist* ΔA, in the same inducible system. The new *Xist* mutants allowed us to study the function of F, B, and C repeats, previously reported to be important for PRC1/PRC2 recruitment at initiation of XCI [10,20]. We show here that a *Xist* ΔC-inducible mutant has no obvious defect in *Xist* RNA coating of the X chromosome and PcG recruitment. This contrasts with previous findings suggesting a role for the C repeat in *Xist* localization [50], although this could be due to differences in the cell type examined (somatic cells versus inducible ES cells), and the technology used (locked nucleic acids—LNAs versus genetic deletions) to destabilize the C repeat. We also show that our *Xist*-inducible mutants lacking the B repeat have impaired PRC1 and PRC2

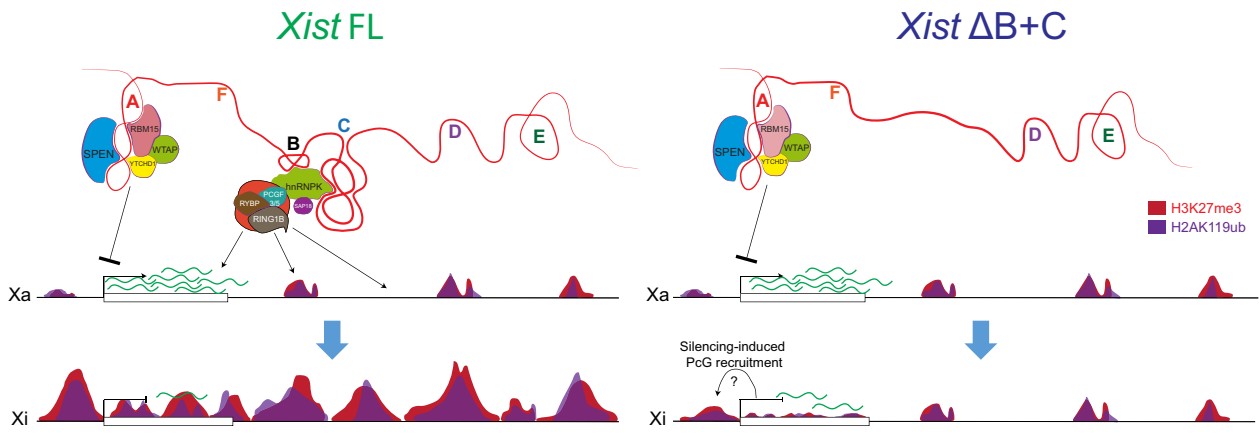

**Figure 5. Working model for *Xist*-mediated PcG recruitment influence on transcriptional silencing based on the phenotypes of the *Xist* ΔB+C mutant.**

SPEN and proteins of the m6A RNA methylation machinery interact with the A repeat to initiate X-linked gene silencing; PCGF3/5-PRC1 recruitment via hnRNPK interaction with the B and C repeats is responsible for the accumulation of H2AK119ub and concomitant enrichment of the PRC2-mark H3K27me3 over the entire X chromosome. In the absence of B and C repeats, there is no enrichment of PcG marks in intergenic regions, but a slight increase at silencing X-linked genes is seen; this could be caused by passive recruitment induced by gene silencing; nevertheless, recruitment of these marks is necessary to stabilize the initial silencing mediated by the A repeat interactors.

recruitment to the X chromosome, consistent with the recent findings implicating a 600-bp region containing the B repeat (and 3 out of 14 motifs of the C repeat) in an autosomal context [15] or the B repeat alone from the endogenous locus in the context of transformed tetraploid MEFs and late differentiating ES cells (day 14) [14]. However, we found that lack of B repeat alone is unable to fully compromise PRC1/PRC2 recruitment on the X chromosome in our inducible *Xist* system (Fig 1C). In the case of Pintacuda *et al* (2017) study, the differences might arise from the fact that *Xist* has been DOX-induced from an autosome instead of the X chromosome. In the case of the Colognori *et al* (2019) study, differences might arise from the fact that the Xi in MEFs is less enriched in PcG marks (e.g., H3K27me3) [51]. Moreover, this ΔB mutant causes defects in *Xist* coating which would exacerbate the phenotype, a phenomenon that seems to be overridden in our inducible system based on RNA FISH measurements performed in *Xist* FL and *Xist* ΔB+C-induced cells (Fig EV2D and E).

In our inducible mutants, complete absence of PRC1/PRC2 recruitment as judged by IF/RNA FISH is seen only if both B and C repeats are deleted. Interestingly, B and C repeats correspond precisely to the binding sites of the RNA-binding protein hnRNPK as mapped by iCLIP: B repeat represents the stronger binding region for hnRNPK within *Xist*, but this protein also interacts all along the C repeat [52]. hnRNPK was recently proposed to be an important player in mediating *Xist*-dependent recruitment of PCGF3/5-PRC1 and PRC2 to the X chromosome [13–15]. In accordance with this, we found that hnRNPK, alongside with non-canonical PRC1 members, is lost from the *Xist* ΔB+C protein interactome as revealed by ChIRP-MS. Interestingly, SAP18 was another protein lacking from the *Xist* ΔB+C interactome (Fig 2C; Dataset EV1). The potential role for this histone deacetylase complex subunit in XCI remains to be addressed. However, SAP18 is unlikely to affect PcG recruitment. Indeed, recent knockdown experiments showed that SAP18 does not affect SMCHD1 recruitment to the Xi, which is a secondary event to H2AK119ub deposition [53].

The *Xist* ΔB+C mutant cannot bind PCGF3/5-PRC1 (Fig 2B and C) or induce global PRC1/PRC2 recruitment to the X chromosome

(Figs 1B and C, and 3A). This provide us with a model to appreciate the role of *Xist*-mediated PcG recruitment on the initiation of transcriptional silencing without affecting pre-marked PcG sites on the X chromosome prior *Xist* expression [9,51] or the other PcG functions throughout the genome. Interestingly, *Xist* ΔB+C mutant is able to cause chromosome-wide transcriptional silencing in contrast to the silencing-defective *Xist* ΔA mutant (Figs 4C–E and EV5D) expressed under the same inducible promoter. This can be explained, at least in part, by the interaction of *Xist* ΔB+C RNA with factors involved in X-linked gene silencing, such as SPEN, RBM15, and WTAP [13,32,35] (Fig 2B and C). However, a slight relaxation of transcription silencing is still seen for the *Xist* ΔB+C mutant RNA (Figs 4C–E and EV5D). It is a weaker phenotype than the previously reported decrease in transcriptional silencing seen in *Xist* DOX-inducible transgenes on autosomes in the context of *Pcgf3*−/− *Pcgf5*−/− double mutant [10] or in PcG-defective *Xist* mutant [15] in ES cells. However, the overall decrease in *Xist*-mediated gene silencing efficiency in an autosomal context [24,42] might render autosomal genes more susceptible to PcG loss. Recently, a PcG-defective *Xist* ΔB mutant also showed a reduction in transcriptional silencing at day 14 of differentiation, although the defects could be partially explained by inefficient *Xist* spreading along the X chromosome and are due to maintenance function of PcG rather than initiation [14]. In any case, in all these systems, it is clear that abrogation of PcG recruitment will result in some degree of relaxation of transcriptional silencing which, nevertheless, does not match the nearly complete loss of *Xist*-mediated silencing in the absence of the A repeat (Figs 4C–E and EV5D) [15,30,38]. Relaxation of transcriptional silencing upon *Xist*-mediated PcG recruitment was also reported by an article published during the revision period of this manuscript [54]. The extent of relaxation was stronger than what we report here, but again was not comparable to the loss of SPEN or *Xist*'s A repeat [54]. In line with this, a recent study using a tiling CRISPR approach to induce indels along a *Xist*-inducible transgene on an autosome was only able to underline the A repeat as the main silencing domain of *Xist* [55]. Overall, the slight relaxation of

transcriptional silencing seen in our *Xist* ΔB+C-inducible mutant suggests that *Xist*-dependent global PcG recruitment to the X chromosome, which is not directed specifically to genes, will be important to guarantee a stable inactive state.

Our nChIP-seq data confirmed the lack of global enrichment of PcG marks of the X chromosome, but revealed a mild enrichment of these marks around the promoters and gene bodies of some X-linked genes in *Xist* ΔB+C expressing cells. This suggests that PcG marks may be laid down on the X chromosome in more than one way. One possibility is that another region of *Xist* mediates PcG recruitment to these genes. Although the A repeat has been previously implicated in PRC2 recruitment [56,57], the specificity of such an interaction is unclear [58,59]. Furthermore, the PRC2 core components have not been identified to bind to *Xist* RNA in different proteomic searches of the *Xist* interactome [13,32,60]. Another possibility is that low levels of PcG proteins may simply be recruited to promoters and gene bodies as a consequence of gene silencing following *Xist* ΔB+C RNA coating. Taking advantage of comparable nChIP-seq and RNA-seq datasets in *Xist* FL and *Xist* ΔB+C, we detected multiple genes which have been silenced and yet with negligible accumulation of H3K27m3 or H2AK119ub at their promoter regions in *Xist* ΔB+C-induced cells (Fig 4F). This suggests that *Xist*-mediated gene silencing can occur in the absence of PcG recruitment, at least, for a subset of X-linked genes. The other subset of X-linked genes has detectable residual accumulation of PcG marks, which could play a role in the initiation of their silencing. Alternatively, the recruitment of PcG at X-linked genes could be secondary to transcriptional silencing in PCGF3/5-PRC1-unbound *Xist* ΔB+C. It has been proposed that PcG recruitment to active promoters and gene bodies could be passive upon their silencing, in the sense that the PcG system will operate on any transcriptionally inactive, GC-rich locus [47–49] (Fig 5). Interestingly, our results with *Xist* ΔB+C also clearly indicate that transcriptional silencing is not sufficient to recruit PcG on X-linked genes to the same extent as inducible *Xist* FL. Thus, lncRNA-directed PcG recruitment, which mechanistically might differ from the passive recruitment to silencing genes, is necessary for proper PcG targeting in the context of XCI.

In conclusion, our results reinforce the idea that *Xist* is a multi-tasking RNA molecule with several structural and regulatory modules [31] that have different functions. We also show that initiation of *Xist*-mediated transcriptional silencing can occur in the absence of *Xist*-mediated PcG recruitment, at least, for a subset of X-linked genes. Our work places *Xist*-mediated PcG recruitment as an important player during XCI needed to sustain initiation of PcG-independent gene silencing.

# Materials and Methods

## Cell lines

The previously published *Xist*-TetOP (herein *Xist* FL) and *Xist*ΔSX-tetOP (herein *Xist* ΔA) XY ES cells [30] were adapted and maintained in feeder-free classic ES cell medium—DMEM media containing 15% fetal bovine serum (FBS), $10^3$ U/ml leukemia inhibitor factor (LIF), $10^{-4}$ mM 2-mercaptoethanol, 50 U/ml penicillin, and 50 μg/ml of streptomycin (Gibco). *Xist* FL ES cell line was used to generate several *Xist* mutants: ΔF+B+C, ΔF+B, ΔB+C, ΔB+1/2C,

ΔB, and ΔC (see Generation of *Xist*-TetOP mutants by CRISPR-Cas9 genome editing).

All ES cells were grown at 37°C in 8% $CO_2$, and medium was changed daily. Inducible expression of *Xist* driven by a TetO promoter was achieved by adding DOX (1.5 μg/ml), while differentiating the ES cells in LIF withdrawal medium—DMEM media containing 10% FBS, $10^{-4}$ mM 2-mercaptoethanol, 50 U/ml penicillin, and 50 μg/ml of streptomycin (Gibco), for 2–5 days, depending on the experiment.

Female MEFs were grown in DMEM media containing 10% FBS, $10^{-4}$ mM 2-mercaptoethanol, 50 U/ml penicillin, and 50 μg/ml of streptomycin (Gibco) at 37°C in 5% $CO_2$.

## Generation of *Xist*-TetOP mutants by CRISPR/Cas9 genome editing

To generate *Xist*-TetOP mutants, $4 \times 10^5$ cells were co-electroporated with 2.5 μg each of two pX459 plasmids (Addgene) expressing the Cas9 endonuclease and chimeric guide RNAs (gRNAs) flanking the region to delete using a Neon Transfection System (Thermo Fisher Scientific). The sequences of the gRNAs used to generate each type of *Xist* mutant are shown in Appendix Table S1. To pick single ES cell clones containing the desired mutations, ES cells were separated by limiting dilution. As soon as visible, single colonies were picked under a microscope and screened for deletion by PCR and absence of the wild-type band with the primers depicted in Appendix Table S1. Positive clones of each *Xist* mutant type were expanded and further validated for the mutation and absence of the wild-type band. Amplicons from the deletion PCR were gel-purified using NZYGelpure kit (NZYTech) and sequenced by the Sanger method (GATC—Eurofins Genomics) (Fig EV1A; Table EV1) using either the forward or the reverse primers (Appendix Table S1). PCR across the deleted regions within *Xist* exon 1 was also performed and confirmed in cDNA obtained upon 4 days of differentiation in DOX conditions, while no band (or very faint bands) was obtained in noDOX conditions (Fig EV1A). *Xist*-TetOP mutants were also analyzed for expression in DOX and noDOX conditions, using primers across exon 1 to exon 3, upstream of the B repeat, and downstream of the C repeat using the primers in Appendix Table S2, and presence or absence of the expected band was in accordance with the respective mutant analyzed (Fig EV1B).

## RT–PCR analysis

Total RNA was isolated from the different *Xist*-TetOP mutant ES cells at D4 of differentiation (from both DOX and noDOX conditions) using NYZol (NZYtech) and then DNase I-treated (Roche) to remove contaminating DNA. RNA template was reverse-transcribed using the Transcriptor High Fidelity cDNA Synthesis Kit (Roche), according to the manufacturer's instructions. cDNA was subjected to RT–PCR using the deletion primers in Appendix Table S1 for the respective *Xist*-TetOP mutants and for the analyses in Fig EV1B using the primers in Appendix Table S2 for all the *Xist*-TetOP mutants.

## RNA polymerase II inhibition

Around $2 \times 10^5$ *Xist* FL and *Xist* ΔB+C ES cells were plated on a well of a six-well plate 72 h prior to actinomycin D treatment. The next

day, differentiation was initiated in DOX conditions and 48 h after cells were treated with 5 μg/ml actinomycin D (Sigma, #A1410) for 2, 4, and 6 h. Total RNA was collected at each time point, including induced cells without treatment (0 hrs), from three independent biological replicates. cDNA synthesis (Roche) was performed after DNase I treatment of RNA samples. Percentage of *Xist* remaining upon treatment was quantified by RT–qPCR and presented as the ratio of *Xist* transcripts at the time of collection relative to *Xist* transcripts in non-treated cells measured using *Xist* exon 1–exon 3 primers normalized to 18S rRNA (Appendix Table S2), which is mainly transcribed by the RNA polymerase I.

## RNA FISH

RNA FISH probes for *Xist* (a 19 kb genomic λ clone 510 [61], *Pgk1* (a 15–16 kb genomic sequence starting 1.6 kb upstream of *Pgk1* gene up to its intron 6) (kind gift from T. Nesterova, Univ. of Oxford) [33], *Lamp2* (RP24-173A8 bacterial artificial chromosome—BAC), and *Rlim/Rnf12* (RP24-240J16 BAC—BACPAC Resources Center) were prepared using the Nick Translation Kit (Abbot) with red and/or green dUTPs (Enzo Life Sciences). RNA FISH was done accordingly to established protocols [61] in *Xist*-TetOP mutant differentiating ES cells with minor modifications. Briefly, cells were dissociated with trypsin (Gibco) and adsorbed onto poly-L-lysine (SIGMA)-coated 22 × 22 mm coverslips for 5 min. Cells were then fixed in 3% paraformaldehyde (PFA) in phosphate-buffered saline (PBS) for 10 min at room temperature (RT) and permeabilized with 0.5% Triton X-100 diluted in PBS with 2 mM vanadyl–ribonucleoside complex (VRC; New England Biolabs) for 5 min on ice. Coverslips were then washed twice in ethanol (EtOH) 70% for 5 min and then dehydrated through ethanol series (80%, 95%, and 100%) and air-dried quickly before hybridization with the fluorescent-labeled probes. Probes were ethanol-precipitated with sonicated salmon sperm DNA (and mouse *Cot1* DNA for *Lamp2* and *Rlim/Rnf12* probes), denatured at 75°C for 7 min (in the case of *Lamp2* and *Rlim/Rnf12* BAC probes, they were let incubating at 37°C for 30 min after denaturation to allow *Cot1* DNA to bind to the repetitive DNA present in these BACs to prevent non-specific hybridization). *Xist* (in red or green) and one X-linked gene (*Pgk1*, *Lamp2*, or *Rlim/Rnf12*) (in green when *Xist* probe was red; in red when *Xist* probe was green) probes were co-hybridized in FISH hybridization solution (50% formamide, 20% dextran sulfate, 2× SSC, 1 μg/μl BSA, 10 mM vanadyl–ribonucleoside) overnight. Washes were carried out with 50% formamide/2× saline-sodium citrate (SSC), three times for 7 min at 42°C, and then with only 2× SCC, three times for 5 min at 42°C. After the RNA FISH procedure, nuclei were stained with 4′,6-diamidino-2-phenylindole (DAPI; Sigma-Aldrich), diluted 1:5,000 in 2× SCC for 5 min at RT, and mounted with Vectashield Mounting Medium (Vectorlabs). Cells were observed with the widefield fluorescence microscope Zeiss Axio Observer (Carl Zeiss MicroImaging) with 63× oil objective using the filter sets FS43HE, FS38HE, and FS49. Digital images were analyzed with the FIJI platform (https://fiji.sc/). To determine the number of cells with an *Xist*-coated X chromosome, a minimum of 250 cells were counted per single experiment. To determine the expression of the different X-linked genes studied (*Pgk1*, *Lamp2*, and *Rlim/Rnf12*), at least 59 cells with a *Xist*-coated X chromosome were counted in DOX conditions, and at least 100 cells were counted in noDOX conditions per experiment.

## Stellaris RNA FISH

Stellaris RNA FISH to measure the size and intensity of *Xist* foci was performed in *Xist* FL and *Xist* ΔB+C ES cells differentiated for 2 days in DOX conditions and in female MEFs on gelatin-coated 22 × 22 mm coverslips. We designed two sets of Stellaris RNA FISH probes using the Stellaris™ Probe Designer software (Biosearch Technologies) for two regions of *Xist* (exon 1 3′ end and exon 7). Each set comprised 48 singly labeled oligonucleotides labeled with Quasar® 570 dye (Appendix Table S3). Hybridization conditions for RNA FISH were followed according to Stellaris™ guidelines using a final concentration of 125 nM of each probe set per coverslip. Briefly, cells were washed with PBS and fixed with 3.7% PFA in PBS for 10 min at RT. After rinsing with PBS and washed one time with 70% EtOH, samples were incubated with 70% EtOH for 1 h at RT. Then, samples were washed in washing buffer (10% formamide/2× SSC) for 5–10 min at RT before probe hybridization. The coverslips containing the samples were then removed from the washing buffer and transferred to parafilm containing 25 μl of hybridization buffer (10% dextran sulfate/10% formamide/2× SSC) with 125 nM of each probe set per coverslip and incubated overnight at 37°C in a moist chamber. The following days, cells were washed twice with washing buffer (30 min at 37°C), followed by a single wash with 2× SSC (5 min at RT). After, nuclei were stained with DAPI (Sigma-Aldrich), diluted 1:10,000 in 2× SCC for 5 min at RT, followed by two washes in 2× SSC (5 min at RT), before being mounted with Vectashield Mounting Medium (Vectorlabs).

Z-stack images (40 slices at 0.4 μm) of each sample were acquired in a Zeiss Cell Observer fluorescence widefield microscope (Carl Zeiss Microimaging) equipped with an Axiocam 506 mono CCD camera using a 63×/1.4 Plan-Apochromat objective and filter sets FS49 for DAPI and FS43HE for Quasar 570. The acquired z-stacks were deconvolved using the Huygens Remote Manager software (Scientific Volume Imaging, The Netherlands, http://svi.nl), using the CMLE algorithm, with SNR:50 and 100 iterations. Deconvolved z-stacks were then processed and analyzed in FIJI (https://fiji.sc/). Briefly, maximum-intensity projections were calculated for each z-stack, and after threshold segmentation, the area (in μm²) and total intensity (area × mean intensity) of each *Xist* foci were measured. At least 71 *Xist* foci signals were quantified from 6 to 7 images obtained from two independent experiments (at least three images per biological replicate). Statistically significant differences between samples were determined using unpaired Student's *t*-test.

## IF/RNA FISH

IF/RNA FISH experiments were performed as previously [20]. *Xist* FL and mutant ES cells were differentiated for 48 h in the presence of DOX (1.5 μg/ml) on gelatin-coated 22 × 22 mm coverslips. Cells were fixed in 3% PFA in PBS for 10 min at RT, followed by permeabilization in PBS containing 0.5% Triton X-100 and VRC (New England Biolabs) on ice for 5 min. After three rapid washes in PBS, samples were blocked for, at least, 15 min with 5% gelatin from cold water fish skin (Sigma) in PBS. Coverslips were incubated with the following primary antibodies diluted in blocking solution at the desired concentration (H3K27me3—Active Motif #39155 1:200; H2AK119ub—Cell Signaling #8240 1:200; JARID2—Abcam #ab48137 1:500; RING1B—Cell Signaling #5694 1:100; EZH2—Leica

Microsystems #NCL-L-EZH2 1:200) in the presence of a ribonuclease inhibitor (0.8 µl/ml; Euromedex) for 45 min at RT (in the case of RING1B antibody, incubation lasted for 4 h). After three washes with PBS for 5 min, the coverslips were incubated with a secondary antibody (goat anti-mouse or anti-rabbit antibodies conjugated with Alexa green, red, or Cy5 fluorophores diluted 1:500) for 45 min in blocking solution supplemented with ribonuclease inhibitor (0.8 µl/ml; Euromedex). Coverslips were then washed three times with PBS for 5 min at RT. Afterward, cells were postfixed with 3% PFA in PBS for 10 min at RT and rinsed three times in PBS and twice in 2× SSC. Excess of 2× SSC was removed, and cells were hybridized with a Xist p510 probe labeled with Alexa green or red dUTPs (prepared and hybridized as mentioned in the RNA FISH protocol). After the RNA FISH procedure, nuclei were stained with DAPI (Sigma-Aldrich), diluted 1:5,000 in 2× SCC for 5 min at RT, and mounted with Vectashield Mounting Medium (Vectorlabs). Cells were observed with the widefield fluorescence microscope Zeiss Axio Observer (Carl Zeiss MicroImaging) with 63× oil objective using the filter sets FS43HE, FS38HE, FS50, and FS49. Digital images were analyzed with the FIJI platform (https://fiji.sc/). Enrichment of the different histone marks or PcG proteins fluorescent signals over Xist cloud marked by RNA FISH was counted from at least 50 cells per single experiment.

### Xist ChIRP-MS

Xist FL (both in DOX and in noDOX conditions) and Xist ΔB+C (DOX) cells were differentiated for 3 days. A fraction of these cells were used to quantify levels of Xist induction by RNA FISH. Xist ChIRP-MS was conducted using a previously published protocol [13] with the following modifications: (i) Around 500 million cells per ChIRP-MS experiment were collected (roughly 10–15 15-cm$^2$ dishes) cross-linked in 3% formaldehyde for 30 min, followed by 0.125 M glycine quenching for 5 min; (ii) all 100 mg of cell pellets was then dissolved in 1 ml of nuclear lysis buffer (50 mM Tris–HCl pH 7.0, 10 mM EDTA, 1% SDS), and 880 µl was sonicated in a Covaris ultrasonicator for 1 h (20 min, three times). Clarified lysates were pooled for each sample; (iii) instead of RNase treatment, noDOX condition was used as control. 6 µl of RiboLock RNase inhibitor was added per ml of clear lysate into the experiment, and control tubes were incubated at 37°C for 30 min prior to hybridization step. Final protein samples were size-separated in three parts from Bis–Tris SDS–PAGE gel and sent for LC/MS-MS. Correct retrieval of Xist RNA after ChIRP from Xist FL and Xist ΔB+C was analyzed by RT–qPCR using three pairs of primers along Xist RNA [Pair 1—forward 1 (Fw1): GCCT CTGA TTTA GCCA GCAC, reverse 1 (Rv1): GCAA CCCA GCAA TAGT CAT; Pair2—Fw2: GACA CAAA TGGG AGCT GGTT, Rv2: GGAT CCTG CACT GGAT GAGT; and Pair 3—Fw3: GCCA TCCT CCCT ACCT CAGAA; Rv3: CCTG ACAT TGTT TTCC CCCT AA) and Gapdh as housekeeping gene (Fw: AAGG TCAT CCCA GAGC TGAA; Rv: CTGC TTCA CCAC CTTC TTGA)]. For details on ChIRP probe design, see Extended Experimental Procedure on the previously published protocol [13].

Xist hits from ChIRP-MS were ranked according to Xist FL DOX/Xist FL noDOX fold change in peptide counts. To calculate this and Xist ΔB+C/Xist FL noDOX ratios, when peptide counts for Xist FL noDOX samples were 0, it was considered 1 (Dataset EV1). For comparison with Chu et al list [13], only annotated protein isoforms with an Annotation score in UniProtKB ≥ 3 (out of 5) were considered with a minimum of 2.5 DOX/noDOX fold change in one of the samples. Proteins present in the Chu's list with DOX/no DOX ratio inferior in Xist ΔB+C than Xist FL were considered underrepresented in Xist ΔB+C protein interactome (Fig 2C). Proteins with no peptide counts for Xist ΔB+C or with equal peptide counts to Xist FL noDOX, which had a Xist FL DOX/noDOX ratio ≥ 4, were considered not to be part of the Xist ΔB+C protein interactome (Fig 2C).

### nChIP-seq

nChIP-seq was performed in duplicates for Xist ΔB+C ES cells at day 2 of differentiation upon DOX and noDOX conditions and compared to results previously obtained for Xist FL [9]. The protocol was followed as described [9]. Briefly, around 3.5 million cells were used per immunoprecipitation (IP) experiment. A fraction of these cells was always used to quantify levels of Xist induction by RNA FISH. Ten million cells were resuspended and lysed in 90 µl of lysis buffer (50 mM Tris–HCl, pH 7.5; 150 mM NaCl; 0.1% sodium deoxycholate; 1% Triton X-100; 5 mM CaCl$_2$; protease inhibitor cocktail; 5 mM sodium butyrate) for 10 min on ice. Lysis buffer with MNase (62 µl) was then added for chromatin digestion and incubated at 37°C for exactly 10 min. Then, 20 mM EGTA was added to stop the reaction, followed by 15,871 g centrifugation for 5 min at 4°C to sediment undigested debris. Supernatant was then transferred, and equal amount of STOP buffer (50 mM Tris–HCl, pH 7.5; 150 mM NaCl; 0.1% sodium deoxycholate; 1% Triton X-100; 30 mM EGTA; 30 mM EDTA; protease inhibitor cocktail; 5 mM sodium butyrate) was added to the samples kept on ice.

Lysate (5 µl) was mixed with 45 µl of proteinase K (ProtK) digestion buffer (20 mM HEPES; 1 mM EDTA; 0.5% SDS) and incubated at 56°C for 30 min. AMPure XP beads (50 µl) were mixed with the digested lysate with 60 µl of 20% PEG8000 1.25 M NaCl for 15 min at RT. Beads were separated on a magnet and washed twice with 80% EtOH for 30 s. DNA was eluted in 12 µl of low-EDTA TE and measured using Qubit DNA High-Sensitivity Kit to normalize lysate concentration between samples. DNA isolated in this step was used for the input sample. The volume of each undigested lysate was adjusted for equal concentration to obtain 1 ml per IP using a 1:1 mix of Lysis Buffer and STOP Buffer.

Protein-A Dynabeads (10 µl/IP) were washed twice in blocking buffer (0.5% BSA; 0.5% Tween in PBS) before being resuspended in blocking buffer and coated with H3K27me3 [1 µg/IP] (Cell Signaling, Cat#9733S) or H2AK119ub [0.4 µg/IP] (Cell Signaling, Cat# 8240) antibodies for 4 h at 4°C. Once coated beads were magnet-separated and resuspended in 1 ml of concentration-adjusted lysate, samples were left rotating overnight at 4°C.

In the following day, beads were magnet-separated and washed quickly with ice-cold washing buffers with low salt buffer (0.1% SDS; 1% Triton X-100; 2 mM EDTA; 20 mM Tris–HCl, pH 8.1; 150 mM NaCl; 0.1% sodium deoxycholate). IPs were then washed four times with low salt buffer, twice with high salt buffer (0.1% SDS; 1% Triton X-100; 2 mM EDTA; 20 mM Tris–HCl, pH 8.1; 360 mM NaCl; 0.1% sodium deoxycholate), and twice with LiCl buffer (0.25 M LiCl; 1% NP40; 1.1% sodium deoxycholate; 1 mM EDTA; 10 mM Tris–HCl pH 8.1). Prior to elution, all samples were rinsed once in TE. ChIP-DNA was eluted in ProtK digestion buffer by incubating at 56°C for 15 min. Beads were separated, and the

supernatant was further digested for 2 h more at 56°C. DNA was isolated using AMPure XP beads as described for the input sample.

For each nChIP-seq, 0.5 μl of input samples was also used to verify the digestion efficiency using D1000 tapestation. Remaining DNA concentration was adjusted and used for library preparation using Ovation® Ultralow Library System V2 following suppliers protocol. Amplified libraries were size-selected for dinucleotide fraction (350- to 600-bp fragments) using agarose gel separation and MinElute Gel Extraction Kit (Qiagen). Sample quality was inspected using D1000 tapestation. Samples were sequenced with HiSeq 2500 using single-end 50-bp mode.

Adapters and low-quality bases (< Q20) have been removed from the sequencing data with TrimGalore (v0.4.0; http://www.bioinformatics.babraham.ac.uk/projects/trim_galore) and Cutadapt (1.8.2) [62]. Reads were then mapped to the mm10 genome with Bowtie2 (2.2.5) with options [–end-to-end -N1 -q] [63]. Duplicates were discarded with Picard MarkDuplicates (1.65) with options [REMOVE_DUPLICATES = true] [64], reads mapped with low quality (< q10) were removed using samtools (1.3) [65], and reads mapped on blacklisted regions from Encode Consortium [66] were discarded. Bigwig files were created with bedtools genomeCoverageBed (2.25.0) [67], using a scale factor calculated on the total library (10,000,000/total reads) and loaded on UCSC genome browser [68].

ChIP-seq signal was then analyzed per window. A global analysis was first done on fixed windows (10 kb) spanning the whole genome and then on different genomic subcategories: initially active gene bodies, initially active promoters, and intergenic regions. Initially active genes were defined in our previous study [9] as genes with a transcript having its TSS (refFlat annotation [69] overlapping a peak of H3K27ac in noDOX samples). For genes having several active transcripts detected, the active gene was defined as starting at the minimum start of transcripts and ending at the maximum end of transcripts. This way, 6,096 initially active genes were defined genome-wide, 286 being on the X chromosome. The initially active gene bodies were defined as those initially active genes excluding the 2 first Kb downstream from TSS. Initially active promoters were defined as ± 2 kb windows around the TSS of these genes. Intergenic regions were defined as 10 kb windows not overlapping a gene (active or inactive) and its promoter (2 kb downstream) or a peak of H3K27ac [9]. Reads overlapping defined windows were then counted with featureCounts (1.5.1) [70] with default options.

For global analysis, count normalization was performed based on counts falling in autosomal consensus peaks. For each histone mark, peaks were first identified in each sample using MACS2 [71], with options [–broad -B -broad-cutoff 0.01] and with input as control, and only peaks with a minimum fold change of 3 were selected. Then, consensus peaks were defined as common regions between peaks identified in a minimum of 2 among the 4 noDOX samples using bedtools multiIntersectBed (2.25.0) and bedtools merge (2.25.0) [67]. For each sample, a normalization factor was calculated with the trimmed mean of M-value method (TMM) from edgeR package [72], based on reads overlapping consensus peaks located on autosomes. To correct for chromatin accessibility or mappability bias, 10-kb windows with outliers counts in the input (counts superior or inferior to mean ± 1.5 SD) were discarded from the analysis. Moreover, to represent read accumulation between DOX and noDOX conditions, normalized initial

counts from noDOX samples were subtracted to corresponding DOX normalized counts.

For genomic subcategories analysis (active gene bodies, active promoters, and intergenic regions), windows that had less than one read per 50 bp for more than 2 among the 8 samples were removed for the analysis. Normalization factors were calculated based on windows located on autosomes, with TMM method using edgeR [72]. Linear regression was then fitted for each window according to the following model: Y = clones + clones:condition, with Y being the log2(cpm) and condition being noDOX or DOX, using Voom function from Limma R package [73]. Significance of coefficients was assessed by a moderated t-statistics, and P-values were corrected by Benjamini–Hochberg procedure. Because of the high variability in proportion of cells with Xist induction, we quantified the number of cells with a Xist cloud by RNA FISH experiments: Xist FL DOX#1—46.64%, Xist FL DOX#2—59.44%, Xist ΔB+C DOX#1—66.30%, and Xist ΔB+C DOX#2—56.29%. Linear regression including the percentage of induction calculated by RNA FISH was also fitted for each window according to the following model: ~0 + clones + clones:induction, using Voom function from Limma R package [73]. The slope of this regression represents then the logFC between noDOX and DOX conditions if the induction of the cell population was complete (corrected logFC).

Metaplots were created using DeepTools (3.0.2) [74]. Bigwigs of log2(FC) between DOX and noDOX samples were first created with personalized scaling according to normalized factors calculated above for active promoters using DeepTools bamCompare. Then, bigwigs of mean of log2(FC) between replicates were then created using DeepTools bigwigCompare with options [–binSize 100 –operation mean], matrix counts were then generated using DeepTools computeMatrix around TSS of active gene coordinates (see above) on the X chromosome and autosomes separately, and plots were then created using DeepTools PlotProfile.

### RNA-seq

Duplicates samples of Xist FL, Xist ΔA and Xist ΔB+C ES cells were differentiated until day 2 in DOX and noDOX conditions. Total RNA was isolated using NYZol (NZYTech) and then DNase I-treated (Roche) to remove contaminating DNA following the manufacturer's recommendations. Initial RNA quality was checked by electrophoresis and sent to NOVOGENE for sequencing. Quality of the samples was verified on a 2100 Agilent Bioanalyser system, and only samples with RIN score above 9 were processed. RNA (1 μg) was used for 250–300 bp insert cDNA library following manufacturer's recommendations (except for Xist ΔB+C DOX#2, which only 100 ng of RNA was used for the library preparation using a low input method). Libraries were sequenced with NovaSeq 6000 platform using paired-end 150-bp mode.

Reads were mapped on mm10 genome with Tophat (2.1.0) [75], with options [-g 1 -x 1 -N 5 –read-edit-dist 5 –no-coverage-search], with refFlat annotation [69]. Reads covering exons of each gene were then counted with featureCounts (1.5.1) with options [-C -p] [70]. Bigwig files were created with Deeptools bamCoverage (2.2.4) [74], with option [–normalizeUsingRPKM], and loaded on UCSC genome browser [68].

Clustering of samples based on normalized counts of X chromosome (calculated with cpm function from edgeR) was done with

hclust function with parameter [method = "Ward.D"], using Pearson correlation as distance.

Differential analysis was done on genes for which 6 among the 12 samples have a TPM superior to 1. Count normalization was done based on counts falling in expressed autosomal genes, with the trimmed mean of M-value method (TMM) from edgeR package [72]. Such as for nChIP-seq analysis, linear regression was then fitted for each gene with models including DOX/noDOX information, or percentage of induction calculated by RNA FISH (*Xist* FL DOX#1 – 46.6%, *Xist* FL DOX#2 – 51.7%, *Xist* ΔA DOX#1 – 53.2%, *Xist* ΔA DOX #2 – 54.5%, *Xist* ΔB+C DOX#1 – 61.4%, *Xist* ΔB+C DOX#2 – 56.7%).

Expression and nChIP-seq data were integrated as following: First, for each mutant analyzed in both sets of data (*Xist* FL and *Xist* ΔB+C), four groups of genes were defined based on log2(FC) of RNA-seq: [−∞, −1.5], [−1.5, −1], [−1, −0.5], and [−0.5, ∞]. For each of these groups, the accumulation of normalized reads from nChIP-seq data at promoters of corresponding genes (DOX-noDOX signal, for genes with enough coverage (see nChIP-seq part above)), was extracted, for each histone mark separately (H3K27me3, H2AK119ub). For each group of genes, the normalized nChIP-seq read enrichment relative to noDOX between both cell lines was then compared using a Wilcoxon test (Fig EV5F).

For each mark, promoters were divided into 2 categories: the ones with no accumulation or residual accumulation of H3K27me3 and H2AK119ub marks, and the ones with substantial accumulation of these repressive marks. The threshold between those two categories was defined as the mean + SD of normalized signal accumulation between DOX and noDOX conditions (normalized reads DOX - normalized reads noDOX) on autosomes in *Xist* ΔB+C samples. Based on the data, the thresholds were 224 for H3K27me3 and 100 for H2AK119ub. Then, two categories of initially active promoters were defined based on both repressive marks: the ones with no or little accumulation for any of the two repressive marks (H3K27me3; H2AK119ub) and the ones with substantial accumulation of one or both repressive marks (Fig EV5G). The CpG content of each category was calculating using bedtools (2.25.0) with options "nuc – pattern G" and options "nuc –pattern CG", and both were compared using a Wilcoxon test (Fig EV5H). Then, for each cell line (*Xist* FL, *Xist* ΔB+C), the expression log2(FC) of the genes not accumulating (or accumulating residual marks) and genes accumulating repressive marks in *Xist* ΔB+C were compared using a Wilcoxon test. Moreover, inside the same category, expression log2(FC) was compared between *Xist* FL and *Xist* ΔB+C cell lines using a paired Wilcoxon test (Fig 4F).

## Statistics

Statistical analysis used for each experiment is indicated in the respective figure legend with *P*-values indicated or marked as \*\**P*-value < 0.05, \*\*\**P*-value < 0.01. Student's *t*-tests compared to control or untreated conditions were used to analyze IF/RNA FISH (Figs 1C and EV2B), actinomycin D treatment (Fig EV2C), and RNA FISH data (Figs 4A and B, and EV2C and D). For analysis of ChIP-seq and RNA-seq data, Wilcoxon tests were used (Figs 3B and 4E and F, and EV4C–E, and EV5D–H). To determine differential expressed genes, Limma analysis were performed (Figs 4D and EV5E). Clustering of samples based on normalized counts of the

X chromosome (calculated with cpm function from edgeR) was done with hclust function with parameter [method = "Ward.D"], using Pearson correlation as distance (Fig 4C).

## Data availability

The datasets produced in this study are available in the following databases:

- nChIP-seq and RNA-seq data: Gene Expression Omnibus GSE123743 (https://www.ncbi.nlm.nih.gov/geo/query/acc.cgi?acc = GSE123743).
- ChIRP-MS Proteomic data: PRIDE PXD014287 or https://doi.org/10.6019/PXD014287 (https://www.ebi.ac.uk/pride/archive/projects/PXD014287).

**Expanded View** for this article is available online.

## Acknowledgements

We thank Elphège Nora, Rafael Galupa, Martin Escamilla-Del-Arenal, and Sérgio de Almeida for critical reading of the manuscript. We are also grateful to members of the Heard Lab (Curie Institute), members of the Carmo-Fonseca's Lab (iMM JLA), in particular to Kenny Rebelo, and also to Pierre Gestraud and Nicolas Servant (U900, Curie Institute) and Aurélie Teissandier (U934/UMR3215, Curie Institute) for their help and critical input to this project. We also want to thank the Bioimaging facility, in particular José Rino, at iMM JLA for their technical support with fluorescent light microscopy and imaging analysis. This work was supported by Fundação para a Ciência e Tecnologia (FCT), project grants PTDC/BEX-BCM/2612/2014 (A.C.R. and S.T.d.R), PTDC/BIA-MOL/29320/2017 IC&DT (C.G. and S.T.d.R), and IF/00242/2014 (V.P. and S.T.d.R), by an ERC Advanced Investigator award ERC-ADG-2014 671027 attributed to E.H. and Sir Henry Wellcome Postdoctoral Fellowship (J.J.Z.), by the Scleroderma Research Foundation (Y.Q., H.Y.C) and by US National Institutes of Health NIH P50-HG007735 (H.Y.C.). H.Y.C. is an Investigator of the Howard Hughes Medical Institute. Publication costs were supported by UID/BIM/50005/2019, project funded by Fundação para a Ciência e a Tecnologia (FCT)/ Ministério da Ciência, Tecnologia e Ensino Superior (MCTES) through Fundos do Orçamento de Estado.

## Author contributions

Conception and/or design of the investigation: AB, JJZ, EH, and STR; methodology: AB, ACR, JJZ, HYC, EH, and STR; data collection: AB, ACR, JJZ, CP, VBP, YQ, CG, and STdR; data analysis and interpretation: AB, ACR, JJZ, LS, and STdR; writing–original draft: STR; writing–review and editing: AB, ACR, JJZ, HYC, and EH; supervision: HYC, EH, and STR; funding acquisition: HYC, EH, and STR.

## Conflict of interest

H.Y.C. is a co-founder of Accent Therapeutics, Boundless Bio, and advisor to 10X Genomics, Arsenal Biosciences, and Spring Discovery. The other authors declare that they have no conflict of interest.

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
