## [Review Process File · EMBO Reports]

The role of *Xist*-mediated Polycomb recruitment in the initiation of X-chromosome inactivation

Aurélie Bousard, Ana Cláudia Raposo, Jan Jakub Žylicz, Christel Picard, Vanessa Borges Pires, Yanyan Qi, Cláudia Gil, Laurène Syx, Howard Y. Chang, Edith Heard, Simão Teixeira da Rocha

Review timeline:

Submission date:	28 February 2019
Editorial Decision:	5 March 2019
Revision received:	19 June 2019
Editorial Decision:	19 July 2019
Revision received:	29 July 2019
Accepted:	6 August 2019

Editor: Achim Breiling

Transaction Report:

1st Editorial Decision

5 March 2019

Thank you for the submission of your manuscript, the referee reports from your previous submission (to a journal outside EMBO press with whom we have a portable peer review agreement), and your revision plan to our editorial offices. I now read your manuscript, and went through the referee reports and your letter.

All referees acknowledge the potential interest of the findings. Nevertheless, all three referees have raised a number of concerns and suggestions to improve the manuscript, or to strengthen the data and the conclusions drawn. Looking at the reports, I feel that a significantly revised manuscript could be suitable for publication at EMBO reports, provided the referee concerns are adequately addressed. I will not detail the reports again, but I think all points need to be addressed, either experimentally, or in a point-by-point response. In particular, I think that point 4 of referee #1 (XY male ES cell line used), point 1 of referee #2 (novelty of findings and discussion of contrasting results), and points 2 and 3 of referee #3 need particular attention.

Given the constructive referee comments, we would like to invite you to revise your manuscript with the understanding that all referee concerns must be fully addressed in the revised manuscript and in a complete point-by-point response. Acceptance of your manuscript will depend on a positive outcome of a second round of review using the same referees that have already assessed the study (their identity has been revealed to as due to the portable peer review agreement with the other journal), provided they agree to look into the study again. It is EMBO reports policy to allow a single round of revision only and acceptance or rejection of the manuscript will therefore depend on the completeness of your responses included in the next, final version of the manuscript.

Revised manuscripts should be submitted within three months of a request for revision; they will otherwise be treated as new submissions. Please contact us if a 3-months time frame is not sufficient for the revisions so that we can discuss the revisions further.

Please refer to our guidelines for preparing your revised manuscript:
<http://embor.embopress.org/authorguide#manuscriptpreparation>

See also our guide for figure preparation:
http://www.embopress.org/sites/default/files/EMBOPress_Figure_Guidelines_061115.pdf

Please also format the references according to EMBO reports style. See:
<http://embor.embopress.org/authorguide#referencesformat>

Supplementary/additional data: The Expanded View format, which will be displayed in the main HTML of the paper in a collapsible format, has replaced the Supplementary information. You can submit up to 5 images as Expanded View. Please follow the nomenclature Figure EV1, Figure EV2 etc. The figure legend for these should be included in the main manuscript document file in a section called Expanded View Figure Legends after the main Figure Legends section. Additional Supplementary material should be supplied as a single pdf labeled Appendix. The Appendix includes a table of content on the first page, all figures and their legends. Please follow the nomenclature Appendix Figure Sx throughout the text and also label the figures according to this nomenclature. For more details please refer to our guide to authors.

Important: All materials and methods should be included in the main manuscript file.

Regarding data quantification and statistics, can you please specify the number "n" for how many experiments were performed, the bars and error bars (e.g. SEM, SD) and the test used to calculate p-values in the respective figure legends? This information must be provided in the figure legends. Please provide statistical testing where applicable. See:
<http://embor.embopress.org/authorguide#statisticalanalysis>

We now strongly encourage the publication of original source data with the aim of making primary data more accessible and transparent to the reader. The source data will be published in a separate source data file online along with the accepted manuscript and will be linked to the relevant figure. If you would like to use this opportunity, please submit the source data (for example scans of entire gels or blots, data points of graphs in an excel sheet, additional images, etc.) of your key experiments together with the revised manuscript. Please include size markers for scans of entire gels, label the scans with figure and panel number, and send one PDF file per figure.

- a complete author checklist, which you can download from our author guidelines (<http://embor.embopress.org/authorguide#revision>). Please insert page numbers in the checklist to indicate where the requested information can be found.
- a letter detailing your responses to the referee comments in Word format (.doc)
- a Microsoft Word file (.doc) of the revised manuscript text
- editable TIFF or EPS-formatted single figure files in high resolution (for main figures and EV figures)

I look forward to seeing a revised version of your manuscript when it is ready. Please let me know if you have questions or comments regarding the revision.

REFEREE REPORTS

Referee #1:

Bousard et al. mutagenize the endogenous Xist locus to determine the role of specific conserved repetitive elements within the Xist RNA in the recruitment of Polycomb proteins and the histone modifications they deposit on the inactive X- chromosome. Published work does not agree as to whether the PRC2 or PRC1 Polycomb complexes are required for X-inactivation. By abrogating binding of the Polycomb proteins to the Xist RNA through deletions of specific repeats, the authors surmised they could test the requirement of the Polycomb group in silencing genes on the inactive-

X. The authors therefore engineer the native *Xist* locus in an XY male embryonic stem cell (ESC) line to inducibly express the full-length as well as various mutants in which the conserved repeats are deleted alone or, in some instances, together. The authors found the B and C repeats within exon 1 of *Xist* RNA were together required to cytologically enrich PRC1 and PRC2 components as well as their attendant histone marks (H2AK119ub and H3K27me3) to the inactive-X. By ChIP, however, in the *Xist* DB+C mutant cells some residual H2AK119ub as well as some H3K27me3 are present at promoters and at gene bodies of X-linked genes. Despite the dramatically reduced H2AK119ub and H3K27me3, X-linked genes are nevertheless largely appropriately silenced. Below are a set of major and minor comments on the manuscript.

Major comments:

1. To test the impact of the Polycomb group proteins on X-inactivation, the most direct route is to ablate the proteins themselves and then test if X-inactivation is perturbed. The advantage of this alternate approach is in fact highlighted by the finding that there is residual H2AK119Ub and H3K27me3 marks on the inactive-X in the *Xist* DB+C mutant cells. The presence of the low amount of these marks cannot be excluded as the reason that there isn't a significant defect in X-linked gene silencing in the *Xist* DB+C mutant cells.

2. Previous work by the Magnuson lab has demonstrated that PRC2 function is required for the maintenance of imprinted X-inactivation and dispensable for the initiation of random X-inactivation (Wang et al., *Nature Genetics* 2001; Kalantry et al., *Nature Cell Biology* 2006; Kalantry and Magnuson, *PLoS Genetics* 2006). To determine if PRC proteins are required in X-inactivation initiation and/or maintenance in ESCs, two tests are necessary. First, it is necessary to profile ESCs as X-inactivation is initiating, by differentiating the ESCs into epiblast-like cells or EpiLCs, as the Heard group has previously done. To test if PRC complexes are required to maintain X-inactivation, a time course of ESC or EpiLC differentiation is required. The cells in the study are assayed at only a single time point: day 2 or day 3 of ESC differentiation while being exposed to Doxycycline (to induce the *Xist* RNA in the engineered XY male ESCs). At these time points, X-inactivation initiation may already have occurred but maintenance hasn't progressed far enough along to see a defect in X-linked gene silencing.

3. Using the ChIRP technique pioneered by Howard Chang's lab, the authors purify *Xist* RNA binding proteins from a mouse ES cell line expressing the full length *Xist* RNA (*Xist* FL) as well as from another ES cell line expressing the *Xist* DB+C RNA. The control *Xist* FL RNA binds RING1B protein, which is an enzymatic subunit of the Polycomb PRC1 complex. The *Xist* DB+C RNA expressing cell line, however, lacks such binding, implicating the B and C repeats as important in RING1B binding. A number of other *Xist* RNA binding proteins, however, are in fact much more highly enriched in the *Xist* DB+C sample compared to the *Xist* FL sample. The authors state that this higher enrichment is due to the increased number of *Xist* RNA coated cells in the *Xist* DB+C cell line compared to the *Xist* FL cell line. But, the proportion of cells with *Xist* RNA coats is 45 +/- 6% for the *Xist* FL cell line and 60 +/- 8% for the *Xist* DB+C cell line. This difference between the two genotypes appears insufficient to explain the disproportionate enrichment in the *Xist* DB+C cell line of a number of *Xist* RNA bound proteins, such as SPEN, RBM15, RNF20, RNF40, and HNRNPC. The increased interaction of these proteins with the *Xist* DB+C RNA could, in principle, compensate for the absence of RING1B interaction and mask any silencing defect due to RING1B absence in the *Xist* DB+C cells. Conversely, the *Xist* DB+C RNA also displays absence of interaction with SAP18, a histone deacetylase complex subunit. Thus, it is not possible to ascribe the slight deficit in silencing that the authors do parse out only to the PRC1 complex.

4. The authors use an XY male ES cell line to modify the *Xist* locus and place it under the control of an Dox-inducible promoter. Is the expression of the induced *Xist* RNA expression similar to that found in differentiating XX female ES cells? The implication being to what extent is the lack of X-linked silencing defect in the *Xist* DB+C mutant cells due to excess *Xist* RNA expression in these cell lines? Such a super-physiological level of *Xist* RNA expression may in fact underlie the increased frequency of ESCs that exhibit *Xist* RNA coating in the *Xist* DB+C mutant cells.

5. A recent paper in *Science* concluded that PCGF3/5-PRC1 complex is required for X-inactivation in ES cells and in mouse embryos (Almeida et al., 2017). This finding is in contrast with the current study, which does not find significantly defective X-linked gene silencing upon deletion of the B+C repeats, which are responsible for binding the PRC1 enzyme RING1B.

Minor Comments:

1. In the introduction, the authors state that there is a maintenance defect in X- inactivation in extra-embryonic tissues of female mouse embryos with a hypomorph mutation in the PRC2 gene Eed. This mutation is considered a null mutation, if I am not mistaken. Work by Montgomery et al. (2005; Current Biology), from the Magnuson lab, demonstrated that this Eed mutation results in the loss of all H3K27 methylation states, consistent with the mutation functioning as a null mutation.

Reviewer #2:

This manuscript presents a very thorough analysis of Polycomb recruitment by Xist, and its role in silencing of an X chromosome. The authors undertake deletions of Xist at its endogenous location; but driven by an inducible promoter. Such an approach monitors the impact from the endogenous location; however, it is analysing the impact in male cells under an inducible promoter. I would have appreciated more description of the inducible promoter, and the transcript levels relative to what is observed in a female.

The conclusions of the study (as presented in the abstract) are that in the context of the X chromosome there is a role for the C repeats, and that silencing can be induced in the absence of Polycomb recruitment. Neither observation is novel; however, there has been considerable controversy around the topic. The previously identified minimal Polycomb recruitment region (XR-PID, Pintacuda et al, 2017) included some C repeats as well as the B repeat region, quite in line with the presented results. The recently published Zylicz et al paper supports the conjecture that silencing occurs without Polycomb (is that the same paper that is called in press?), as did an earlier study by Wutz et al. (2002), with analysis of K27me3 by Kohlmaier et al 2004 (perhaps better reference to cite than da Rocha?). However, Almeida et al. (2017) also showed loss of silencing upon deletion of Pcgf3/5.

(1) I felt there should be more discussion of why the different systems would yield contrasting results.

(2) The variability between cells expressing Xist was quite dramatic, and surprising with an inducible system. While mentioned, the text comments on the full-length being 45 +/- 6% rather than the dramatically lower 24% discussed on p7. Can Xist be induced in differentiated cells (which would address whether the difference was the proportion of differentiated cells in the population)?

(3) The text states that d2 was chosen as it is the maxima of Polycomb recruitment. Is this also true for this system, and what happens beyond the maxima with respect to polycomb and the marks they establish (H3K27me3; H2AK119Ub)?

(4) For the ChIRP results, I would have liked to see more details of the comparison to the Chu et al list. In the figure the Chu et al ranking could be included in the green box. A table listing the proteins identified (as well as those from Chu et al) would have been helpful (the source Data did not appear to include such analysed data, only the raw data). There appear to be 'hits' found in delBC, that are not found in the FL??

Furthermore, how were some proteins chosen to be shown in orange - are these significant? It would be helpful to highlight a window of significance, and then the significance of the hits seen the 'reverse way' could be assessed. Was JARID identified in the screen?

Minor Comments:

(1) Figure 1 - in order to clarify that the 3' domain remains, perhaps the terminal line in the 'blow up' of the deletions could be 'squiggly' or dotted.

(2) In Figure 3 it is difficult to see arrows for transcription orientation and this could be remedied by an additional arrow below the gene name.

(3) Fig 4, S1, panel E; nothing appears significant so there are no red dots (but this presentation leaves the audience looking for red dots); what are the dashed lines if nothing is significant?

(4) There were some grammatical issues regarding tense being used.

Additional data files and statistical comments:
These comments appear in body of review.

Reviewer #3:

This manuscript by the Teixeira da Rocha and Heard and their respective co-workers concerns the epigenetic mechanism of X chromosome inactivation in mammals. The

presented cell-based study was performed on mouse embryonic stem (ES) cells and focused on the long non-coding RNA Xist, which brings about X-inactivation in cis on the X chromosome after its expression becomes randomly upregulated during the very early steps of ES cell differentiation. The aim of the study was to determine the role in PcG protein recruitment and X-linked gene inactivation of different parts of the Xist RNA using an elegant inducible CRISPR-Cas9 based deletion system. The main finding of this study is that domains B and C are essential for PRC recruitment and for H3K27me3 and H2AK119ub establishment on the inactive chromosome, but that this domain is not strictly required for the initiation of inactivation at genes. This interesting finding is complemented with detailed 'CHIRP-MS' studies which confirm the importance of the B+C domains of Xist RNA for several key proteins (including RNF2/RING1B, PCGF5, SAP18 and hnRNPK, which is essential for recruitment of a non-canonical PRC1 complex).

Overall, the study is clearly presented and the experiments are of high quality and conclusive, and nicely complement recent similar studies on different parts of the mouse Xist RNA. However, several aspects could have been developed better, to make this study of higher interest. This includes a lack of information on whether B+C-lacking Xist, besides initiating X-inactivation in the in vitro model, is also sufficient for the maintenance of the X linked gene repression, beyond the initial few days of ES differentiation. Another aspect is the reduced size of the Xist RNA foci and how this could be explained, and whether the B+C-deleted RNA would have a comparable signal stability at the Xi as the full length RNA.

Major points:

- 1) As concerns the B+C deletion, the authors should show its effects upon ES- differentiation, beyond the initial 2-4 days. This important point can be readily addressed in the in vitro system used. Linked to this question, the Materials and Methods should provide dedicated information on how ES cell differentiation was performed.
- 2) The authors conclude that the RNA FISH foci for the F+B and F+B+C RNAs are smaller than for the full-length Xist RNA. Can this be quantified? Additionally, how stable maintained are these foci, and the ones detected for delta-B+C, upon RNA polymerase-2 inhibition. In other words, do the truncated forms show a similar maintained (stable) presence on the X chromosome as the full-length Xist RNA?
- 3) It remains rather confusing what controls the PRC complex recruitment and the observed H3K27me3 and H2AK119ub at the 'active gene promoter's in the B+C- deleted Xist cells (Figure 3). The authors propose that this is because of these genes being inactive (last line page 9), but in the different figures they indicate these genes as being 'active'. In case the PCG protein recruitment here would be a default mechanism of recruitment to lowly transcribed genes, not dependent on Xist, it should be relevant to compare the observed levels of H3K27me3 and H2AK119ub with those at inactive autosomal genes as well.

Minor comments:

-On page 8 it is mentioned that PCGF3 was lacking in the delta-B+C protein interactome in the CHIPP-MS experiment. However, this protein is not apparent in Figure 2B or 2C (is only PCGF5 strongly affected?). Please provide more information in the text about this finding on PCGF3 and its significance.

-Please explain better what is meant with 'PCGF3/5-PRC1 complex'. Generally, provide more information about the Xist-associated proteins that are considered in the study. This is already for some, but many proteins are not explained in enough detail.

- In figure 1A, please put the map of Xist RNA and the presentation of the different deletion at the same scale.
- The title is rather non-informative and does not convey the main conclusions of this study.

Additional data files and statistical comments:

The supplementary information seems sufficient. I am not knowledgeable enough to judge the statistical information.

1st Revision – authors response

19 June 2019

Reviewer #1 (General assessment and major comments (Required)):

Bousard et al. mutagenize the endogenous Xist locus to determine the role of specific conserved repetitive elements within the Xist RNA in the recruitment of Polycomb proteins and the histone modifications they deposit on the inactive X-chromosome. Published work does not agree as to whether the PRC2 or PRC1 Polycomb complexes are required for X-inactivation. By abrogating binding of the Polycomb proteins to the Xist RNA through deletions of specific repeats, the authors surmised they could test the requirement of the Polycomb group in silencing genes on the inactive-X. The authors therefore engineer the native Xist locus in an XY male embryonic stem cell (ESC) line to inducibly express the full-length as well as various mutants in which the conserved repeats are deleted alone or, in some instances, together. The authors found the B and C repeats within exon 1 of Xist RNA were together required to cytologically enrich PRC1 and PRC2 components as well as their attendant histone marks (H2AK119ub and H3K27me3) to the inactive-X. By ChIP, however, in the Xist $\Delta B+C$ mutant cells some residual H2AK119ub as well as some H3K27me3 are present at promoters and at gene bodies of X-linked genes. Despite the dramatically reduced H2AK119ub and H3K27me3, X-linked genes are nevertheless largely appropriately silenced. Below are a set of major and minor comments on the manuscript.

Major comments:

1. To test the impact of the Polycomb group proteins on X-inactivation, the most direct route is to ablate the proteins themselves and then test if X-inactivation is perturbed. The advantage of this alternate approach is in fact highlighted by the finding that there is residual H2AK119Ub and H3K27me3 marks on the inactive-X in the Xist $\Delta B+C$ mutant cells. The presence of the low amount of these marks cannot be excluded as the reason that there isn't a significant defect in X-linked gene silencing in the Xist $\Delta B+C$ mutant cells.

We thank the reviewer to raise these concerns. The aim of this manuscript is to decipher the role of Xist-mediated recruitment of Polycomb proteins (PcG) in the initiation of transcriptional silencing of the X chromosome. The usage of KO approaches will deplete H3K27me3 and/or H2AK119ub marks everywhere in the genome, including on the X chromosome prior to Xist expression. PcG deletion delays ESC differentiation (da Rocha et al., 2014; Leeb et al., 2010; Pasini et al., 2007), a necessary precondition for Xist induction and XCI. Hence PcG inactivation poses substantial complications when testing the role of Polycomb proteins in XCI. We wanted to establish a system where we could disrupt Xist-dependent recruitment of Polycomb proteins, without affecting pre-marked PcG sites prior Xist induction on the X chromosome, which play a role in Xist-mediated PcG spreading (Pinter et al., 2012; Zyllicz et al., 2019) or the other PRC1/PRC2 functions throughout the genome. Therefore, our Xist $\Delta B+C$ provides an improved model to only disrupt the recruitment of PcG proteins mediated by Xist. We clarified this in the Discussion by adding the following sentence “This provide us with a model to appreciate the role of Xist-mediated PcG recruitment on the initiation of transcriptional silencing without affecting existent pre-marked PcG X-linked sites prior Xist expression (Pinter et al., 2012; Zyllicz et al., 2019) or the other PcG functions throughout the genome.”

In the context of the Xist $\Delta B+C$ mutant cells, we could find X-linked genes undergoing silencing and not yet exhibiting significant accumulation of PcG marks, arguing that X-linked gene silencing can be independent of PcG recruitment at least for a subset of X-linked genes (Fig. 4F-G). Our previous work (Zyllicz et al., 2019) on chromatin dynamics at the initiation of

Xist-induced XCI showed that PcG marks are initially deposit intergenically and subsequently at genes undergoing inactivation. Therefore, in the context of *Xist* $\Delta B+C$ mutant, the residual recruitment of Polycomb marks found in many genes is more likely to be caused by the ensuing transcriptional silencing (Fig. 3B-D; Fig. 4C-F; Fig. EV5G). Indeed, passive PcG recruitment to silenced CpG-enriched promoters has been reported by several groups (Davidovich et al., 2013; Mendenhall et al., 2010; Riising et al., 2014).

Importantly, we take into consideration the comment from the reviewer and acknowledge that our choice of title might not have been the best to precise the main aim of this study which is the dissection of the role of *Xist*-mediated Polycomb recruitment at the initiation stages of XCI. Therefore, we changed the title from: “*Exploring the role of Polycomb recruitment in Xist-mediated silencing of the X chromosome in ES cells*” to “*The role of Xist-mediated Polycomb recruitment in the initiation of X-chromosome inactivation*”.

2. Previous work by the Magnuson lab has demonstrated that PRC2 function is required for the maintenance of imprinted X-inactivation and dispensable for the initiation of random X-inactivation (Wang et al., Nature Genetics 2001; Kalantry et al., Nature Cell Biology 2006; Kalantry and Magnuson, PLoS Genetics 2006). To determine if PRC proteins are required in X-inactivation initiation and/or maintenance in ESCs, two tests are necessary. First, it is necessary to profile ESCs as X-inactivation is initiating, by differentiating the ESCs into epiblast-like cells or EpiLCs, as the Heard group has previously done. To test if PRC complexes are required to maintain X-inactivation, a time course of ESC or EpiLC differentiation is required. The cells in the study are assayed at only a single time point: day 2 or day 3 of ESC differentiation while being exposed to Doxycycline (to induce the *Xist* RNA in the engineered XY male ESCs). At these time points, X-inactivation initiation may already have occurred but maintenance hasn't progressed far enough along to see a defect in X-linked gene silencing.

In this manuscript, we are only able to study the role of *Xist*-dependent Polycomb recruitment in the initiation stages of XCI and we are careful in the manuscript not to relate any of our findings with the maintenance stage of XCI. The role of *Xist*-mediated Polycomb recruitment on the maintenance stage of random XCI has been recently explored by the Lee's lab (Colognori et al., 2019). Our system based on *Xist* induction of the sole X chromosome in XY ESCs causes major cell lethality, which was already demonstrated by Wutz et al. 2002, precluding any meaningful analysis past day 5 of differentiation (Wutz et al., 2002). We allude to this in the Result section: “*Corroborating the nascent-transcript RNA FISH data, we also noted significant reduction in cell survival upon prolonged DOX induction (≥ 5 days) for *Xist* FL and all the mutants with the exception of *Xist* ΔA RNA (Table EV4). This is consistent with efficient XCI in XY ESCs, resulting in functional nullisomy for the X chromosome, and thus cell death*”. In the case of *Xist* $\Delta B+C$ mutant cells, only about $5.1 \pm 1.4\%$ of cells survived when compared DOX to noDOX conditions at D5. For these reasons, we have clearly state in the title we are looking at the initiation stages of XCI.

Nonetheless, to satisfy the reviewer's comment (and also reviewer 2), we provide data in this Rebuttal letter concerning the enrichment of PRC2-related proteins, JARID2 and EED, in a few of our *Xist* mutants at D4 of differentiation (*Xist* FL, *Xist* $\Delta B+C$ and *Xist* $\Delta F+B$) performed in the beginning of the study (Rebuttal Fig. 1; Rebuttal Figures are at the end of this Rebuttal Letter). The results found are similar to the ones at D2 (Table EV2). Since we are interested in the initiation of XCI, we felt it is unnecessary to include D4 data in the manuscript.

3. Using the ChIRP technique pioneered by Howard Chang's lab, the authors purify *Xist* RNA binding proteins from a mouse ES cell line expressing the full length *Xist* RNA (*Xist* FL) as well as from another ES cell line expressing the *Xist* $\Delta B+C$ RNA. The control *Xist* FL RNA binds RING1B protein, which is an enzymatic subunit of the Polycomb PRC1 complex. The *Xist* $\Delta B+C$ RNA expressing cell line, however, lacks such binding, implicating the B and C repeats as important in RING1B binding.

A number of other *Xist* RNA binding proteins, however, are in fact much more highly enriched in the *Xist* $\Delta B+C$ sample compared to the *Xist* FL sample. The authors state that this higher enrichment is due to the increased number of *Xist* RNA coated cells in the *Xist* $\Delta B+C$ cell line compared to the *Xist* FL cell line. But, the proportion of cells with *Xist* RNA coats is $45 \pm 6\%$ for the *Xist* FL cell line and $60 \pm 8\%$ for the *Xist* $\Delta B+C$ cell line. This difference between the two genotypes appears insufficient to explain the disproportionate enrichment in the *Xist* $\Delta B+C$ cell line

of a number of Xist RNA bound proteins, such as SPEN, RBM15, RNF20, RNF40, and HNRNPC. The increased interaction of these proteins with the Xist $\Delta B+C$ RNA could, in principle, compensate for the absence of RING1B interaction and mask any silencing defect due to RING1B absence in the Xist $\Delta B+C$ cells. Conversely, the Xist $\Delta B+C$ RNA also displays absence of interaction with SAP18, a histone deacetylase complex subunit. Thus, it is not possible to ascribe the slight defect in silencing that the authors do parse out only to the PRC1 complex.

For the ChIRP-MS experiment, the percentage of cells with Xist domains was estimated by RNA FISH as following: Xist $\Delta B+C$ (50.9%) and Xist FL (24.0%) (Figure EV3B) and not $45\pm 6\%$ for the Xist FL cell line and $60\pm 8\%$ as mentioned by the reviewer (this refers to the analysis at D4 on Figure EV1C). Furthermore, the amount of Xist retrieved after ChIRP procedure was remarkably more in Xist $\Delta B+C$ than Xist FL as judged by RT-qPCR for three independent primers (fold enrichment around 2.5 to 4-fold) (Figure EV3A). This justifies the “disproportionate enrichment” of common protein in Xist $\Delta B+C$ versus Xist FL in this experiment which are of the same order of magnitude (2.5 to 3-fold on average). We also provide new data based on single-molecule RNA FISH for Xist using Stellaris oligo-probes to show that Xist coating does not differ between Xist FL and Xist $\Delta B+C$ induced cells (Fig. EV2D-E). Therefore, the “disproportionate enrichment” does not result from differences among the Xist-expressing cells in Xist $\Delta B+C$ and Xist FL, but rather from the higher number of cells with Xist domains in Xist $\Delta B+C$ compared to Xist FL in this experiment. This relevant information is clearly stated in the text on the Result section: “The Xist RNA levels recovered from Xist $\Delta B+C$ were higher than for Xist FL induced cells (Fig EV3A), due to a higher proportion of cells inducing Xist (50.9% vs. 24.0%) (Fig EV4B) but not because of increased RNA induction levels in individual cells (Fig EV2D-E).”

Besides hnRNPK and PCGF3/5-PRC1, another protein not found in the Xist $\Delta B+C$ interactome was SAP18. Although SAP18 was not a top interactor (only ranked 51th among Chu et al. interactors), it could, nonetheless, potentially affect the mild silencing defect in Xist $\Delta B+C$. However, this is unlikely to affect Polycomb recruitment based on recent results (Jansz et al., 2018). For this reason, we added the following passage in our Discussion section: “Interestingly, SAP18, although not a top interactor, was another protein lacking from the Xist $\Delta B+C$ interactome (Fig 3C; Dataset EV1). The potential role for this histone deacetylase complex subunit in XCI remains to be addressed. However, SAP18 is unlikely to affect Polycomb recruitment. Indeed, recent knockdown experiments showed that SAP18 does not affect SMCHD1 recruitment to the Xi, which is a secondary event to H2AK119ub deposition (Jansz et al., 2018)”. In any case, to satisfy the reviewer’s point regarding SAP18 function in XCI, we performed preliminary knock-down (KD) experiments (Rebuttal Figure 2). We successfully KD SAP18 as screened by Western Blot (Rebuttal Figure 2A) and checked the effect on Xist expression and on the silencing of three X-linked genes (*Mecp2*, *Pgk1* and *Rnf12*) in Xist FL induced cells at day 2 of differentiation by RT-qPCR. We could neither detect differences in induced Xist expression or in decreased silencing for any of the three X-linked genes tested compared to scramble RNAi (Rebuttal Fig. 2B-C). We feel that a more comprehensive characterization of the potential role of SAP18, if any, on XCI should deserve consideration out of the scope of this manuscript.

4. The authors use an XY male ES cell line to modify the Xist locus and place it under the control of an Dox-inducible promoter. Is the expression of the induced Xist RNA expression similar to that found in differentiating XX female ES cells? The implication being to what extent is the lack of X-linked silencing defect in the Xist $\Delta B+C$ mutant cells due to excess Xist RNA expression in these cell lines? Such a super-physiological level of Xist RNA expression is may in fact underlie the increased frequency of ESCs that exhibit Xist RNA coating in the Xist $\Delta B+C$ mutant cells.

We would like to clarify the reviewer that we did not “modify the Xist locus and place it under the control of an Dox-inducible promoter”. We have rather used an existing mouse XY male ES line with a DOX-inducible promoter generated by Anton Wutz (Wutz et al., 2002). This system has been widely used and extensively characterized in many publications (Chow et al., 2010; Giorgetti et al., 2016; Kohlmaier et al., 2004; Schoeftner et al., 2006; Wutz et al., 2002; Zyllicz et al., 2019). Other inducible systems to study Xist function have also been used in the field by the different labs (da Rocha et al., 2014; Engreitz et al., 2013; Pintacuda et al., 2017; Wutz and Jaenisch, 2000). The reason behind this is that mutagenizing Xist at its endogenous

locus in female cells often causes skewed X-inactivation towards inactivation of the wild type allele (Hoki et al., 2009; Lv et al., 2016; Senner et al., 2011).

A possible drawback from this inducible systems, as the reviewer has pointed out, is the difference between induced levels among *Xist* FL and *Xist* $\Delta B+C$ induced cells and how they are compared to normal *Xist* levels. For this reason, we used a single-cell approach based on RNA FISH using Stellaris oligo-probes against *Xist* RNA in *Xist* FL and *Xist* $\Delta B+C$ induced cells and also in P_{gk12.1} female ESCs at day 2 of differentiation (the conditions mostly used in this manuscript, including for the RNA-seq and ChIP-seq experiments). For the P_{gk12.1} differentiated ESCs, most of the cells did not show a *Xist* domain or have only incipient *Xist* domains typical of cells in a pre-XCI state or still undergoing random choice of the X chromosome to inactive (Rebuttal Fig. 3). This could not be compared to our inducible *Xist* cells. For this reason, we decide to use female mouse embryonic fibroblasts (MEFs) which represents a cell line with stable normal *Xist* domains. Our results depicted in Fig. EV2D-E show no major differences in the size and intensity of *Xist* clouds between *Xist* FL and *Xist* $\Delta B+C$ RNAs. However, both these RNA FISH signals occupies a larger area and are more intense than the ones seen in female MEFs. This data rules out the argument from the reviewer that “Such a super-physiological level of *Xist* RNA expression is may in fact underlie the increased frequency of ESCs that exhibit *Xist* RNA coating in the *Xist* $\Delta B+C$ mutant cells.”. On the other way, these results seem to suggest that our inducible system results in an increase in *Xist* molecules surrounding the Xi. This phenomenon is mentioned in the manuscript several instances, for example, in the Result section: “This might indicate an increase of *Xist* molecules surrounding the Xi in our inducible system compared to normal *Xist* coating during XCI maintenance, which, nonetheless, does not differ between *Xist* FL and *Xist* $\Delta B+C$ RNAs.” We also take this into consideration in our analysis of the results in this revised version. For this reason, we added the following passage in the Discussions section: “A possible disadvantage is the increase in *Xist* RNA molecules surrounding the X chromosome (Fig EV3D-E) which could mask or attenuate a few of the phenotypes. This has been controlled in this study by the use of the wild-type version (*Xist* FL) and a silencing-defective mutant, *Xist* ΔA , in the same inducible system”. In any case, this does not invalidate our main conclusion which is that transcriptional silencing can be initiated in the absence of recruitment of Polycomb complexes by *Xist* (Fig 4F-G; Fig EV5F).

5. A recent paper in Science concluded that PCGF3/5-PRC1 complex is required for X-inactivation in ES cells and in mouse embryos (Almeida et al., 2017). This finding is in contrast with the current study, which does not find significantly defective X-linked gene silencing upon deletion of the B+C repeats, which are responsible for binding the PRC1 enzyme RING1B.

In our study, we found a modest, but statistically significant, decrease in X-linked gene silencing in *Xist* $\Delta B+C$. Therefore, our results are not in opposition to the ones from Almeida et al. 2017. Still, Almeida et al. suggest a stronger effect than the one we documented here. This might be due to the differences in the approach used. They used *Pcgf3*^{-/-}/*Pcgf5*^{-/-} ESCs with a *Xist* inducible transgene on chromosome 16 (Almeida et al., 2017), while we induced *Xist* from the endogenous location on the X chromosome. Since the capabilities of *Xist* to silence are transgene- and location-specific (Almeida et al., 2017; Loda et al., 2017; Pintacuda et al., 2017; Tang et al., 2010), this might explain the differences in the extent of the effects seen by Almeida et al. (2017) and by us. Moreover, *Pcgf3*^{-/-}/*Pcgf5*^{-/-} ESCs result in reduction of H2AK119ub throughout the genome and, moreover, affects pre-bound fraction of PcG on the X chromosome prior *Xist* expression. This could potentially also explain the differences observed. We added the following sentence to stress the advantages of our system in the Discussion: “The *Xist* $\Delta B+C$ mutant cannot bind PCGF3/5-PRC1 (Fig 2B-C) or induce global PRC1/PRC2 recruitment to the X chromosome (Fig 1B-C; Fig 3A). This provide us with a model to appreciate the role of *Xist*-mediated PcG recruitment on the initiation of transcriptional silencing without affecting existent pre-marked PcG X-linked sites prior *Xist* expression (Pinter et al., 2012; Zylitz et al., 2019) or the other PcG functions throughout the genome.” Concerning *Pcgf3*^{-/-}/*Pcgf5*^{-/-} female embryos, Almeida et al. 2017 only documented absence of PcG marks (H3K27me3 and H2AK119ub) from the Xi, but not what happened to the transcriptional silencing of X-linked genes. In fact, the only female-specific phenotype they report is that “female placentas lack trophoblasts altogether and as a consequence fail to form a labyrinth”, which is similar to the role of EED in maintenance of XCI in extraembryonic

lineages (Kalantry et al., 2006; Wang et al., 2001). The role of PCGF3/5-PRC1 in transcriptional silencing of the Xi in the embryonic lineage still needs to be clarified. In any case, we agree we should discuss this in more detail and added the following passage in the Discussion to acknowledge the potential discrepancies between ours and Almeida's et al (and also Pintacuda et al.) findings: *“However, a slight relaxation of transcription silencing is still seen for the Xist ΔB+C mutant RNA (Fig 4C-E; Fig EV5D). It is a weaker phenotype than the previously reported decrease of transcriptional silencing seen in Xist DOX-inducible transgenes on autosomes in the context of Pcgf3^{-/-} Pcgf5^{-/-} mutant (Almeida et al., 2017) or in PcG-defective Xist mutant (Pintacuda et al., 2017) in ES cells. However, the overall decrease in Xist-mediated gene silencing efficiency in an autosomal context (Loda et al., 2017; Tang et al., 2010) might render autosomal genes more susceptible to PcG loss.”*

Reviewer #1 (Minor Comments):

1. In the introduction, the authors state that there is a maintenance defect in X-inactivation in extra-embryonic tissues of female mouse embryos with a hypomorph mutation in the PRC2 gene Eed. This mutation is considered a null mutation, if I am not mistaken. Work by Montgomery et al. (2005; Current Biology), from the Magnuson lab, demonstrated that this Eed mutation results in the loss of all H3K27 methylation states, consistent with the mutation functioning as a null mutation.

The reviewer is right. This was corrected in the text by removing the word “hypomorph”.

Reviewer #2 (General assessment and major comments (Required)):

This manuscript presents a very thorough analysis of Polycomb recruitment by Xist, and its role in silencing of an X chromosome. The authors undertake deletions of Xist at its endogenous location; but driven by an inducible promoter. Such an approach monitors the impact from the endogenous location; however, it is analysing the impact in male cells under an inducible promoter. I would have appreciated more description of the inducible promoter, and the transcript levels relative to what is observed in a female.

The conclusions of the study (as presented in the abstract) are that in the context of the X chromosome there is a role for the C repeats, and that silencing can be induced in the absence of Polycomb recruitment.

Neither observation is novel; however, there has been considerable controversy around the topic. The previously identified minimal Polycomb recruitment region (XR-PID, Pintacuda et al, 2017) included some C repeats as well as the B repeat region, quite in line with the presented results. The recently published Zylitz et al paper supports the conjecture that silencing occurs without Polycomb (is that the same paper that is called in press?), as did an earlier study by Wutz et al. (2002), with analysis of K27me3 by Kohlmaier et al 2004 (perhaps better reference to cite than da Rocha?). However, Almeida et al. (2017) also showed loss of silencing upon deletion of Pcgf3/5.

- (1) I felt there should be more discussion of why the different systems would yield contrasting results.

We thank the reviewer to consider our work *“a very thorough analysis of Polycomb recruitment by Xist”*. In our manuscript, we address the sequence requirements for Xist-mediated PcG recruitment in the context of the X chromosome and the role of a PcG-defective Xist mutant in transcriptional silencing. We made two novel findings:

- (1) In contrast to Pintacuda et al, 2017, we report that only the total removal of the B and C repeats results in complete loss of Polycomb recruitment in the context of an inducible system; even our Xist ΔB+1/2C (1.4 Kb), which is larger than the XR-PID (0.6 Kb) (Pintacuda et al., 2017), doesn't completely abrogate Polycomb recruitment (Figure 1B-C; Table EV2); therefore, in the context of the X chromosome (but not in autosomes), the region required for Polycomb recruitment needs both the B and C repeats.
- (2) Previously, we have uncovered that H2AK119ub along with histone deacetylation were the first chromatin events associated with transcriptional silencing and that

transcription impeded PcG spreading to active genes based on the use of the *Xist* ΔA silencing-deficient mutant (Zylicz et al., 2019). However, only in this study, by disrupting *Xist*-mediated Polycomb recruitment (using the *Xist* $\Delta B+C$ mutant), we could directly address their effect on transcriptional silencing.

The inducible system that we decided to use to perform our CRISPR/Cas9 deletions has been already established and characterized by A. Wutz's lab (Kohlmaier et al., 2004; Schoeftner et al., 2006; Wutz et al., 2002) and has been widely used by the Wutz's (Kohlmaier et al., 2004; Schoeftner et al., 2006; Wutz et al., 2002) and the Heard's lab (Chow et al., 2010; Giorgetti et al., 2016; Zylicz et al., 2019). We added a note on the Result to acknowledge the broad use of this inducible system: *"This system recapitulates hallmarks of XCI, namely chromosome-wide Xist coating, X-linked gene silencing and heterochromatin formation (Wutz et al., 2002) and has been extensively used in the literature (Chow et al., 2010; Giorgetti et al., 2016; Kohlmaier et al., 2004; Schoeftner et al., 2006; Wutz et al., 2002; Zylicz et al., 2019)."*

To satisfy the reviewers concerns, we have also made amendments in the Discussion section to better explain the contrasting results from the different systems in the literature and ours, as, for example: *"However, a slight relaxation of transcription silencing is still seen for the Xist $\Delta B+C$ mutant RNA (Fig. 4C-E; Fig. EV5D). It is a weaker phenotype than the previously reported decrease of transcriptional silencing seen in Xist DOX-inducible transgenes on autosomes in the context of Pcgf3^{-/-} Pcgf5^{-/-} mutant (Almeida et al., 2017) or in PcG-defective Xist mutant (Pintacuda et al., 2017) in ES cells. However, the overall decrease in Xist-mediated gene silencing efficiency in an autosomal context (Loda et al., 2017; Tang et al., 2010) might render autosomal genes more susceptible to PcG loss. Recently, a PcG-defective Xist ΔB mutant also show a reduction of transcriptional silencing at day 14 of differentiation, although the defects could be partially explained by inefficient Xist spreading along the X chromosome and are due to maintenance function of PcG rather than initiation (Colognori et al., 2019). In any case, in all these systems, it is clear that PcG recruitment will result in some degree of relaxation of transcriptional silencing which, nevertheless, does not match the nearly complete loss of Xist-mediated silencing in the absence of the A-repeat (Fig. 4C-E; Fig. EV5D) (Hoki et al., 2009; Pintacuda et al., 2017; Wutz et al., 2002). In line with this, a recent study using a tiling CRISPR approach to induce indels along a Xist inducible transgene on an autosome was only able to underline the A repeat as the main silencing domain of Xist (Wang et al., 2019)."*

Furthermore, we measure the size and intensity of *Xist* RNA FISH foci in female MEFs and our *Xist* FL and *Xist* $\Delta B+C$ induced male cells by RNA FISH using *Xist* Stellaris oligo-probes not overlapping the mutated region. This analysis summarized in Fig. EV2D-E suggests that this inducible system results in *Xist* RNA foci which are larger and stronger than normal *Xist* foci in somatic cells. This is now documented and carefully taken into consideration in the analysis of the results throughout the manuscript, as, for example, in the Discussion section: *"A possible disadvantage is the increase in Xist RNA molecules surrounding the X chromosome (Fig. EV2D-E) which could mask or attenuate a few of the phenotypes. This has been controlled by the use of the wild-type version (Xist FL) and a silencing-defective mutant, Xist ΔA , in the same inducible system"*

(2) The variability between cells expressing *Xist* was quite dramatic, and surprising with an inducible system. While mentioned, the text comments on the full-length being 45 +/- 6% rather than the dramatically lower 24% discussed on p7. Can *Xist* be induced in differentiated cells (which would address whether the difference was the proportion of differentiated cells in the population)? **The variability of *Xist* induction and the fact that we don't achieve 100% of cells with *Xist* domains is a known fact of this system and has been noticed in the literature several times (da Rocha et al., 2014; Giorgetti et al., 2016; Kohlmaier et al., 2004; Zylicz et al., 2019). Variability is vial-dependent and induction seems to diminish at higher cell passages (data not shown). For this reason, for any of our major experiments (ChIRP-MS, nChIP-seq or RNA-seq experiments), we always confirmed carefully the number of cells presenting *Xist* domains by RNA FISH.**

This system has been used before to induce *Xist* expression in E13.5-E16.5 embryos (Kohlmaier et al., 2004). Therefore, *Xist* expression is clearly induced in differentiated cells. Likewise, we did not see differences in *Xist* induction from day 2 (D2) to day 4 (D4) of differentiation, despite the expected increase in the proportion of differentiated cells from D2 to D4 [D2: for *Xist* FL: ~51% and *Xist* $\Delta B+C$: ~59% (Figure EV4D; Figure EV5C); D4: for *Xist* FL: 45 ± 6% and *Xist* $\Delta B+C$: 60 ± 8% (Figure EV1C)].

(3) The text states that d2 was chosen as it is the maxima of Polycomb recruitment. Is this also true for this system, and what happens beyond the maxima with respect to polycomb and the marks they establish (H3K27me3; H2AK119Ub)?

Based on our previous experience, the recruitment of PRC2 proteins and the co-factor JARID2 to the inactive X-chromosome, reaches its maximum at D2 being almost constant until D4 and then starts to drop during differentiation by LIF withdrawal (da Rocha et al., 2014). Given that PcG marks are highly dependent on continuous *Xist* expression, we don't expect differences in this system.

Nonetheless, to satisfy the reviewer's comment (and also reviewer 1), we provide data in this Rebuttal letter concerning the enrichment of PRC2-related protein, JARID2 and EED, in a few of our *Xist* mutants at D4 of differentiation (*Xist* FL, *Xist* ΔB+C and *Xist* ΔF+B) performed in the beginning of the study (Rebuttal Fig. 1; figures are at the end of this Rebuttal letter). The results found are similar to the ones at D2 (Table EV2). Since we are interested in the initiation of XCI, we felt it is unnecessary to include D4 data in the manuscript.

(4) For the ChIRP results, I would have liked to see more details of the comparison to the Chu et al list. In the figure the Chu et al ranking could be included in the green box. A table listing the proteins identified (as well as those from Chu et al) would have been helpful (the source Data did not appear to include such analysed data, only the raw data). There appear to be 'hits' found in delBC, that are not found in the FL?

In the Dataset EV1, we have appended both analysed and raw data and not only raw data as the reviewer pointed out. In fact, this excel file has three data sheets. Sheet 1 named "Peptide counts - *Xist* hits" is a list of all the reported proteins also found in the original Chu et al. (2015) list. Sheet 2 named "Peptide counts - Filtered" is the analysed data set of all the proteins found in the study filtered based Annotation score from UniprotKB (≥ 3 - out of 5). Sheet 3 represents the full data set. All the hits from the Chu et al. list are clearly identified by color code in the three sheets.

We now include the Chu et al. ranking to satisfy the reviewer's request. However, there is an important experimental difference in our approach compared to the one from Chu et al. (2015). We used the *Xist* FL noDOX as the negative control, while Chu et al. (2015) used RNase A treated cells. For this reason, the ratio DOX/noDOX (our experiment) might not be directly comparable to the ratio DOX/RNase A treatment (Chu et al. list). For this reason, we provide this ranking in Dataset EV1 – spreadsheet 1 with the following note to the legend clarifying the experimental differences between the two studies: "*This protein list is ranked according to the *Xist* FL DOX/noDOX ratio from largest to smallest. For comparison, the Chu et al. 2015 ranking on the same *Xist* FL line is displayed – note that in Chu et al. (2015) experiment, RNase A treatment was used as the negative control, instead of noDOX conditions. Therefore the two rankings should not be considered equivalent.*"

Concerning the "hits" found in *Xist* ΔB+C but not *Xist* FL, this is due to the fact that for this specific experiment, the percentage of cells with *Xist* domains was quite different (*Xist* ΔB+C: 50.9% and *Xist* FL: 24.0%) as measured by RNA-FISH (Figure EV3B). Furthermore, the amount of *Xist* retrieved after ChIRP procedure was higher in *Xist* ΔB+C than *Xist* FL (fold enrichment varies from 2.5-fold for primer pair 3 to 4-fold for primer pair 1) (Figure EV3A). Overall the ChIRP-MS worked better on the *Xist* ΔB+C cells. This justifies why common *Xist* "hits" between *Xist* ΔB+C and *Xist* FL are generally more abundant in the mutant case (in same order of magnitude: 2.5 to 3-fold on average) and also, why a few of the previous *Xist* hits (from Chu's list) found in the same *Xist* FL ES line (Chu et al., 2015) could be found in *Xist* ΔB+C and not in *Xist* FL in this experiment.

Furthermore, how were some proteins chosen to be shown in orange - are these significant? It would be helpful to highlight a window of significance, and then the significance of the hits seen the 'reverse way' could be assessed. Was JARID identified in the screen?

Based on our previous success in mapping A-repeat interactors with relevant roles in XCI as SPEN using a single ChIRP-MS experiment (Chu et al., 2015), we have decided to perform this experiment only once, which hinders the possibility to perform statistical tests.

As mentioned in previous points, retrieved *Xist* RNA from the *Xist* ΔB+C mutant cells were much higher than *Xist* FL (around 2.5-4-fold) (Figure EV3A-B). As a consequence, common interactors were on average 2.5-3-fold more enriched in *Xist* ΔB+C than in *Xist* FL in this

experimental setup. For this reason, we choose a simple criterion for the proteins underrepresented in *Xist* ΔB+C (marked in orange in Fig. 2C) and absent from *Xist* FL (marked in red in Fig. 2C). This choice is explained in the Materials and Methods section on ChIRP-MS: “Proteins present in the Chu’s list with DOX/no DOX ratio inferior in *Xist* ΔB+C than *Xist* FL were considered underrepresented in *Xist* ΔB+C protein interactome (Fig 2C). Proteins with no peptide counts for *Xist* ΔB+C or with equal peptide counts to *Xist* FL noDOX, which had a *Xist* FL DOX/noDOX ratio ≥ 4 were considered not to be part of the *Xist* ΔB+C protein interactome (Fig 2C).” The fact that *Xist* ΔB+C ChIRP-MS worked better than *Xist* FL makes us confident that any hit not found for *Xist* ΔB+C, but enriched in *Xist* FL (present in Chu et al.’s list) is not part of the *Xist* ΔB+C interactome. Our finding that PCGF3/5-PRC1 proteins and hnRNPK do not bind with *Xist* ΔB+C is consistent with our IF/RNA FISH (Fig. 1B-C), nChIP-seq data (Fig. 3A) and with the literature from three independent labs (Cirillo et al., 2016; Colognori et al., 2019; Pintacuda et al., 2017). *Jarid2* was not found in our ChIRP-MS experiment and is not one of 81 proteins found to be part of the *Xist* interactome as defined by Chu et al. (2015).

Reviewer #2 (Minor Comments):

(1) Figure 1 - in order to clarify that the 3' domain remains, perhaps the terminal line in the 'blow up' of the deletions could be 'squiggly' or dotted.

We thank the reviewer to point out for this. We changed Figure 1A to show the deletions in the context of the full RNA (i.e., with the representation of the 3' domains). This was an issue also raised by reviewer 3 which is now solved.

(2) In Figure 3 it is difficult to see arrows for transcription orientation and this could be remedied by an additional arrow below the gene name.

We thank the reviewer to point out for this. This has now been changed in Figure 3D, but also in Figure 4G.

(3) Fig 4, S1, panel E; nothing appears significant so there are no red dots (but this presentation leaves the audience looking for red dots); what are the dashed lines if nothing is significant?

We thank the reviewer for pointing this out. This panel is to show precisely that we could not find statistically significant differences in silencing kinetics for any X-linked genes. We have removed the part of the legend indicating red dots with p-value < 0.5 . The dashed lines are just limits for the 2-fold change and are helpful to highlight the absence of considerable variation between *Xist* FL and *Xist* ΔB+C induced cells.

(4) There were some grammatical issues regarding tense being used.

We have revised the grammatical issues throughout the manuscript.

Reviewer #2 (Additional data files and statistical comments):

These comments appear in body of review.

Reviewer #3 (General assessment and major comments (Required)):

This manuscript by the Teixeira da Rocha and Heard and their respective co-workers concerns the epigenetic mechanism of X chromosome inactivation in mammals. The presented cell-based study was performed on mouse embryonic stem (ES) cells and focused on the long non-coding RNA *Xist*, which brings about X-inactivation in cis on the X chromosome after its expression becomes randomly upregulated during the very early steps of ES cell differentiation. The aim of the study was to determine the role in PcG protein recruitment and X-linked gene inactivation of different parts of the *Xist* RNA using an elegant inducible CRISPR-Cas9 based deletion system. The main finding of this study is that domains B and C are essential for PRC recruitment and for H3K27me3 and H2AK119ub establishment on the inactive chromosome, but that this domain is not strictly required for the initiation of inactivation at genes. This interesting finding is complemented with detailed 'CHIRP-MS' studies which confirm the importance of the B+C domains of *Xist* RNA for several key proteins (including RNF2/RING1B, PCGF5, SAP18 and hnRNPK, which is essential for recruitment of a non-canonical PRC1 complex).

Overall, the study is clearly presented and the experiments are of high quality and conclusive, and nicely complement recent similar studies on different parts of the mouse *Xist* RNA. However,

several aspects could have been developed better, to make this study of higher interest. This includes a lack of information on whether B+C-lacking Xist, besides initiating X-inactivation in the in vitro model, is also sufficient for the maintenance of the X linked gene repression, beyond the initial few days of ES differentiation. Another aspect is the reduced size of the Xist RNA foci and how this could be explained, and whether the B+C-deleted RNA would have a comparable signal stability at the Xi as the full length RNA.

Major points:

-As concerns the B+C deletion, the authors should show its effects upon ES-differentiation, beyond the initial 2-4 days. This important point can be readily addressed in the in vitro system used. Linked to this question, the Materials and Methods should provide dedicated information on how ES cell differentiation was performed.

We thank the reviewer for the overall positive evaluation of the manuscript. The point raised by the reviewer is an interesting one, however, this cannot be addressed with our male ESC inducible system, since this leads to cell death due to X chromosome nullisomy, which hinders any meaningful evaluation past day 5 of differentiation (Wutz et al., 2002). We alluded to this fact on the following passage: “Corroborating the nascent-transcript RNA FISH data, we also noted significant reduction in cell survival upon prolonged DOX induction (≥ 5 days) for Xist FL and all the mutants with the exception of Xist ΔA RNA (Table EV4). This is consistent with efficient XCI in XY ESCs, resulting in functional nullisomy for the X chromosome, and thus cell death”. In the case of Xist $\Delta B+C$ mutant cells, only about $5.1 \pm 1.4\%$ of cells survived when compared DOX to noDOX conditions at D5. For this reason, we are extremely careful in the manuscript not to relate any of our findings with the maintenance stage of XCI. We focused our analysis on the initiation phase of XCI (mostly at day 2 and 3 of differentiation) as we now mention in the title. For the role of different Xist repeats during maintenance stage of XCI, a recent article has just been published (Colognori et al., 2019).

We differentiate the cells by LIF withdrawal. This is clearly explained in the Materials and Methods section: “Inducible expression of Xist driven by a TetO promoter was achieved by adding DOX (1.5 $\mu\text{g/ml}$) while differentiating the ES cells in LIF withdrawal medium - DMEM media containing 10% FBS, 10^{-4} mM 2-mercaptoethanol and 50U/ml penicillin and 50 $\mu\text{g/ml}$ of streptomycin (Gibco), for 2 to 5 days, depending on the experiment.”

-The authors conclude that the RNA FISH foci for the F+B and F+B+C RNAs are smaller than for the full-length Xist RNA. Can this be quantified? Additionally, how stable maintained are these foci, and the ones detected for delta-B+C, upon RNA polymerase-2 inhibition. In other words, do the truncated forms show a similar maintained (stable) presence on the X chromosome as the full-length Xist RNA?

Concerning the size of RNA FISH foci for Xist $\Delta F+B$ and Xist $\Delta F+B+C$, on average, they seem to be smaller as can be appreciated in Figure 1B or Figure EV1C. However, the Xist probe we used for the general characterization of our mutant lines was the classical p510 probe (Chaumeil et al., 2008) which encompasses the whole exon 1 (where all our deletions were performed) up to the beginning of last exon. Therefore, this probe is not the most suitable one to infer differences in Xist RNA FISH foci between mutants. For this reason, we decided to remove the sentence “(on average, Xist $\Delta F+B+C$ and Xist $\Delta F+B$ have smaller domains)” from the manuscript. Since these two Xist mutants are not central to the findings we report in this manuscript (mostly based on the findings of Xist $\Delta B+C$), we believe this point should deserve consideration outside the scope of this manuscript.

Taking into consideration this reviewer’s point nevertheless, we decided to compare and quantify the Xist RNA foci in Xist FL and Xist $\Delta B+C$ induced cells, since alterations in the pattern of those foci might underlie the defects observed in Xist $\Delta B+C$. For that, we developed Stellaris fluorescent oligo-probes against portions of Xist present in both Xist FL and Xist $\Delta B+C$ and measure the size and intensity of RNA FISH signal. Our analysis displayed in Fig. EV2D-E show no dramatic differences between Xist FL and Xist $\Delta B+C$ FISH foci. Therefore, no obvious change in Xist coating could be observed for the Xist $\Delta B+C$ mutant.

The reviewer also raised a question concerning how stably maintained are Xist foci namely for the Xist $\Delta B+C$ RNA mutant. For this reason and as suggested by the reviewer, we performed RNA Polymerase II inhibition using Actinomycin D. Our results revealed no discernable differences in RNA stability between Xist FL and Xist $\Delta B+C$ (Fig. EV2C). Therefore, changes

in the stability of the mutant RNA and on *Xist* coating do not explain its effects on loss of overall PcG recruitment.

-It remains rather confusing what controls the PRC complex recruitment and the observed H3K27me3 and H2AK119ub at the 'active gene promoter's in the B+C-deleted *Xist* cells (Figure 3). The authors propose that this is because of these genes being inactive (last line page 9), but in the different figures they indicate these genes as being 'active'. In case the PCG protein recruitment here would be a default mechanism of recruitment to lowly transcribed genes, not dependent on *Xist*, it should be relevant to compare the observed levels of H3K27me3 and H2AK119ub with those at inactive autosomal genes as well.

We agree with the reviewer that our conclusion may have been somewhat confusing. By “active” genes, we actually meant genes which were initially expressed in non-inducible conditions (noDOX) (based on criteria documented in Zylicz et al., 2019 and stated in the Materials and Methods section), and could then be considered to study *Xist*-dependent enrichment of Polycomb. We have now changed the denomination to “initially active” gene bodies or promoters in the figures, legends and main text. Like this, we will make clear that we are not talking about lowly expressed genes. These are rather initially active genes that became efficiently silenced in *Xist* Δ B+C expressing cells and recruited low levels of PcG marks.

Nonetheless, as suggested by the reviewer, we compared the levels of PcG marks between X-linked genes undergoing silencing in *Xist* Δ B+C, which we believed it is a default mechanism, and on inactive autosomal genes. For that, we generated average plots showing the mean ChIP-seq coverage of H3K27me3 and H2AK119ub at TSS of initially active genes on the X chromosome and at TSS on inactive genes on autosomes in *Xist* Δ B+C cell line upon DOX induction. As a control, we also monitor the same for *Xist* FL. We observed that level of H3K27me3 and H2AK119ub were lower on genes that were silenced during XCI in both *Xist* Δ B+C and *Xist* FL than on genes that are inactive on autosomes (Rebuttal Figure 4; figures are at the end of this Rebuttal Letter). This difference can certainly be explained because we are comparing X-linked genes that are not being silenced in all the cells as some cells do not induce *Xist* expression (Fig. EV4D; Fig. EV5C) with autosomal silenced genes that should be inactive in all the cells. In any case, we see a bigger difference between the X-linked genes and autosomal genes in the PcG-deficient *Xist* Δ B+C than in the *Xist* FL (Rebuttal Figure 4), consistent with our results (Fig. 3B; Fig. EV4E). As this comparison between repressive autosomal genes and genes that are getting inactive on the X-chromosome cannot be perfectly assessed in our system because this does not happen in 100% of cells, we decided not to include this result in the final manuscript.

Reviewer #3 (Minor Comments)

Minor comments - On page 8 it is mentioned that PCGF3 was lacking in the delta-B+C protein interactome in the CHIPP-MS experiment. However, this protein is not apparent in Figure 2B or 2C (is only PCGF5 strongly affected?). Please provide more information in the text about this finding on PCGF3 and its significance.

The Figure 2B is the top 20 hits found in our ChIRP-MS experiment for *Xist* FL, while Fig. 2C is a selection of the hits in *Xist* FL and *Xist* Δ B+C also found in the original *Xist* interactors documented by Chu et al. 2015. PCGF3 was not originally in this list (therefore, it is not on Fig. 2C), but we could nonetheless find it in our experiment to bind to *Xist* FL and not *Xist* Δ B+C. The data on PCGF3 can be found in Dataset EV1. This information is already mentioned carefully in the text: “We also found that PCGF3, which was not in the original Chu et al.’s list, was present in the *Xist* FL interactome, but lacking from the *Xist* Δ B+C interactome in our ChIRP-MS experiment (Dataset EV1).”

-Please explain better what is meant with 'PCGF3/5-PRC1 complex'. Generally, provide more information about the *Xist*-associated proteins that are considered in the study. This is already for some, but many proteins are not explained in enough detail.

We thank the reviewer to bring this to our attention. Indeed, we did not explain properly the meaning of “PCGF3/5-PRC1”. For this reason, we added the following passage in the Introduction to explain the composition and functions of these complexes: “Recently, several studies showed that the non-canonical PCGF3-PRC1 and PCGF5-PRC1 complexes associate with Xist RNA via hnRNPk (Almeida et al., Chu et al., 2015; Colognori et al., 2019; Pintacuda et al., 2017). PCGF3 and PCGF5 are paralogs and both PCGF3-PRC1 and PCGF5-PRC1 complexes are very similar in function (Almeida et al., 2017; Fursova et al., 2019) and composition: besides PCGF3 or PCGF5, they contain the catalytic RING subunits RING1A/RING1B, RYBP/YAF2, FBRS and the accessory proteins CK2 and AUTS2) (Gao et al., 2012; Hauri et al. 2016). For this reason, they will be defined as PCGF3/5-PRC1 throughout this article.”

-In figure 1A, please put the map of Xist RNA and the presentation of the different deletion at the same scale.

We thank the reviewer to point this out, which was also raised by reviewer 2. We changed Figure 1A to show all the deletions of Xist RNA at the same scale.

-The title is rather non-informative and does not convey the main conclusions of this study.

Although we agree with the reviewer, we prefer to maintain the same sort of title with small changes: “The role of Xist-mediated Polycomb recruitment in the initiation of X-chromosome inactivation”. This way, we give the opportunity to the reader to interpret our results on an inducible system in ESCs in the light of normal random X-chromosome inactivation that occurs in female mammals.

Reviewer #3 (Additional data files and statistical comments):

The supplementary information seems sufficient. I am not knowledgeable enough to judge the statistical information.

Rebuttal Figure 1: Recruitment of a PRC2 member (EED) and a PRC2 co-factor (JARID2) in the different *Xist*-TetOP mutants at day 4 of differentiation

Representative images of combined IF for JARID2 (green), EED (grey) with RNA FISH for *Xist* (red) in *Xist*-TetOP lines (for clone 1 of each mutant type) upon D4 in the presence of DOX; DAPI in blue; Numbers within each panel indicates the percentage of cells with JARID2 and EED enrichment on the *Xist*-coated X chromosome (*Xist* FL n=44; *Xist* Δ F+B n=60; *Xist* Δ B+C n=60); The following primary antibodies were used for JARID2 (Abcam ab 48137; diluted 1:500) and EED (diluted 1:100) (Hamer et al., 2002; Sewalt et al., 1998). As secondary antibodies, goat anti-rabbit and anti-mouse conjugated with Alexa 488 (green) Alexa 680 (Cy5) fluorophores were used diluted at 1:500. Images were acquired using a DeltaVision system (Applied Precision) and images were analysed using ImageJ software; Scale bar: 10 μ m.

Rebuttal Figure 2: SAP18 seems not to affect *Xist*-induced transcriptional silencing

A – Western Blot showing the levels of SAP18 upon KD; α -TUBULIN (α -TUB) is used as a loading control; SAP18 antibody - Abcam ab175920; α -TUB antibody - Clone B-5-1-2; Sigma T5168.

B – *Xist* levels quantified by RT-qPCR normalized for *Gapdh* in *Xist* FL cells in scrambled RNAi noDOX, scrambled RNAi DOX and *Sap18* RNAi DOX at day 2 of differentiation; Shown are the mean of three independent biological replicates and error bars represent the standard error of the mean (S.E.M.).

Sequences for primers for *Xist* and *Gapdh* are displayed in Appendix Table S2; *Sap18* KD was achieved using Mouse SMARTpool: siGENOME *Sap18* siRNA (Dharmacon) and compared to siGENOME Non-Targetting siRNA Pool#1 (Dharmacon) as Scrambled control; transfection was achieved using Lipofectamine RNAiMAX (Invitrogen, #13778150).

C – Levels of the X-linked genes *Mecp2*, *Pgk1* and *Rnf12* normalized for *Gapdh* in *Xist* FL cells in scrambled RNAi noDOX, scrambled RNAi DOX and SAP18 RNAi DOX at day 2 of differentiation; Shown are the mean of three independent biological replicates normalized for the Scrambled RNAi noDOX conditions; error bars represent the standard error of the mean (S.E.M.).

For the three X-linked genes, the following primers were used: *Mecp2* - Forward (Fw): TGACTTCACGGTAACTGGGAG; Reverse (Rv): TTTCACCTGAACACCTTCTGATG; *Pgk1* - Fw: CTCCGCTTTCATGTAGAGGAAG; Rv: GACATCTCCTAGTTTGGACAGTG; *Rnf12* - Fw: CCCAGGTGAAAGTACTGAGG; Rv: CTCTCCAGCTCTATTTTCATCG.

Rebuttal Figure 3: Representative deconvolved images from Z-projection images of RNA FISH using Stellaris fluorescent-labelled oligonucleotides targeting *Xist* in PGK12.1 and *Xist* FL induced cells at day 2 of differentiation and MEFs; Scale bar: 10 μ m.

Rebuttal Figure 4: Average plots showing the mean ChIP-seq coverage of H3K27me3 (top) and H2AK119ub (bottom) at TSS of initially active genes on chrX (blue) and at TSS on inactive genes on autosomes (green) in DOX condition. Shown is the normalized coverage based on autosomal signal in both *Xist* FL (left) and *Xist* Δ B+C (right) cell lines.

References:

Almeida, M., Pintacuda, G., Masui, O., Koseki, Y., Gdula, M., Cerase, A., Brown, D., Mould, A., Innocent, C., Nakayama, M., *et al.* (2017). PCGF3/5-PRC1 initiates Polycomb recruitment in X chromosome inactivation. *Science* 356, 1081-1084.

Chaumeil, J., Augui, S., Chow, J.C., and Heard, E. (2008). Combined immunofluorescence, RNA fluorescent in situ hybridization, and DNA fluorescent in situ hybridization to study chromatin changes, transcriptional activity, nuclear organization, and X-chromosome inactivation. *Methods Mol Biol* 463, 297-308.

Chow, J.C., Ciaudo, C., Fazzari, M.J., Mise, N., Servant, N., Glass, J.L., Attreed, M., Avner, P., Wutz, A., Barillot, E., *et al.* (2010). LINE-1 activity in facultative heterochromatin formation during X chromosome inactivation. *Cell* *141*, 956-969.

Chu, C., Zhang, Q.C., da Rocha, S.T., Flynn, R.A., Bharadwaj, M., Calabrese, J.M., Magnuson, T., Heard, E., and Chang, H.Y. (2015). Systematic discovery of Xist RNA binding proteins. *Cell* *161*, 404-416.

Cirillo, D., Blanco, M., Armaos, A., Buness, A., Avner, P., Guttman, M., Cerase, A., and Tartaglia, G.G. (2016). Quantitative predictions of protein interactions with long noncoding RNAs. *Nat Methods* *14*, 5-6.

Colognori, D., Sunwoo, H., Kriz, A.J., Wang, C.Y., and Lee, J.T. (2019). Xist Deletional Analysis Reveals an Interdependency between Xist RNA and Polycomb Complexes for Spreading along the Inactive X. *Molecular cell* *74*, 101-117 e110.

da Rocha, S.T., Boeva, V., Escamilla-Del-Arenal, M., Ancelin, K., Granier, C., Matias, N.R., Sanulli, S., Chow, J., Schulz, E., Picard, C., *et al.* (2014). Jarid2 Is Implicated in the Initial Xist-Induced Targeting of PRC2 to the Inactive X Chromosome. *Molecular cell* *53*, 301-316.

Engreitz, J.M., Pandya-Jones, A., McDonel, P., Shishkin, A., Sirokman, K., Surka, C., Kadri, S., Xing, J., Goren, A., Lander, E.S., *et al.* (2013). The Xist lncRNA exploits three-dimensional genome architecture to spread across the X chromosome. *Science* *341*, 1237973.

Fursova, N.A., Blackledge, N.P., Nakayama, M., Ito, S., Koseki, Y., Farcas, A.M., King, H.W., Koseki, H., and Klose, R.J. (2019). Synergy between Variant PRC1 Complexes Defines Polycomb-Mediated Gene Repression. *Molecular cell* *74*, 1020-1036 e1028.

Giorgetti, L., Lajoie, B.R., Carter, A.C., Attia, M., Zhan, Y., Xu, J., Chen, C.J., Kaplan, N., Chang, H.Y., Heard, E., *et al.* (2016). Structural organization of the inactive X chromosome in the mouse. *Nature* *535*, 575-579.

Hamer, K.M., Sewalt, R.G., den Blaauwen, J.L., Hendrix, T., Satijn, D.P., and Otte, A.P. (2002). A panel of monoclonal antibodies against human polycomb group proteins. *Hybrid Hybridomics* *21*, 245-252.

Hoki, Y., Kimura, N., Kanbayashi, M., Amakawa, Y., Ohhata, T., Sasaki, H., and Sado, T. (2009). A proximal conserved repeat in the Xist gene is essential as a genomic element for X-inactivation in mouse. *Development* *136*, 139-146.

Jansz, N., Keniry, A., Trussart, M., Bildsoe, H., Beck, T., Tonks, I.D., Mould, A.W., Hickey, P., Breslin, K., Iminoff, M., *et al.* (2018). Smchd1 regulates long-range chromatin interactions on the inactive X chromosome and at Hox clusters. *Nat Struct Mol Biol* *25*, 766-777.

Kalantry, S., Mills, K.C., Yee, D., Otte, A.P., Panning, B., and Magnuson, T. (2006). The Polycomb group protein Eed protects the inactive X-chromosome from differentiation-induced reactivation. *Nat Cell Biol* *8*, 195-202.

Kohlmaier, A., Savarese, F., Lachner, M., Martens, J., Jenuwein, T., and Wutz, A. (2004). A chromosomal memory triggered by Xist regulates histone methylation in X inactivation. *PLoS Biol* *2*, E171.

Leeb, M., Pasini, D., Novatchkova, M., Jaritz, M., Helin, K., and Wutz, A. (2010). Polycomb complexes act redundantly to repress genomic repeats and genes. *Genes Dev* *24*, 265-276.

Loda, A., Brandsma, J.H., Vassilev, I., Servant, N., Loos, F., Amirnasr, A., Splinter, E., Barillot, E., Poot, R.A., Heard, E., *et al.* (2017). Genetic and epigenetic features direct differential efficiency of Xist-mediated silencing at X-chromosomal and autosomal locations. *Nat Commun* *8*, 690.

Lv, Q., Yuan, L., Song, Y., Sui, T., Li, Z., and Lai, L. (2016). D-repeat in the XIST gene is required for X chromosome inactivation. *RNA Biol* *13*, 172-176.

Pasini, D., Bracken, A.P., Hansen, J.B., Capillo, M., and Helin, K. (2007). The polycomb group protein Suz12 is required for embryonic stem cell differentiation. *Mol Cell Biol* *27*, 3769-3779.

Pintacuda, G., Wei, G., Roustan, C., Kirmizitas, B.A., Solcan, N., Cerase, A., Castello, A., Mohammed, S., Moindrot, B., Nesterova, T.B., *et al.* (2017). hnRNPK Recruits PCGF3/5-PRC1 to the Xist RNA B-Repeat to Establish Polycomb-Mediated Chromosomal Silencing. *Molecular cell* *68*, 955-969 e910.

Pinter, S.F., Sadreyev, R.I., Yildirim, E., Jeon, Y., Ohsumi, T.K., Borowsky, M., and Lee, J.T. (2012). Spreading of X chromosome inactivation via a hierarchy of defined Polycomb stations. *Genome Res* *22*, 1864-1876.

Schoeftner, S., Sengupta, A.K., Kubicek, S., Mechtler, K., Spahn, L., Koseki, H., Jenuwein, T., and Wutz, A. (2006). Recruitment of PRC1 function at the initiation of X inactivation independent of PRC2 and silencing. *EMBO J* *25*, 3110-3122.

Senner, C.E., Nesterova, T.B., Norton, S., Dewchand, H., Godwin, J., Mak, W., and Brockdorff, N. (2011). Disruption of a conserved region of Xist exon 1 impairs Xist RNA localisation and X-linked gene silencing during random and imprinted X chromosome inactivation. *Development* *138*, 1541-1550.

Sewalt, R.G., van der Vlag, J., Gunster, M.J., Hamer, K.M., den Blaauwen, J.L., Satijn, D.P., Hendrix, T., van Driel, R., and Otte, A.P. (1998). Characterization of interactions between the mammalian polycomb-group proteins Enx1/EZH2 and EED suggests the existence of different mammalian polycomb-group protein complexes. *Mol Cell Biol* *18*, 3586-3595.

Tang, Y.A., Huntley, D., Montana, G., Cerase, A., Nesterova, T.B., and Brockdorff, N. (2010). Efficiency of Xist-mediated silencing on autosomes is linked to chromosomal domain organisation. *Epigenetics Chromatin* *3*, 10.

Wang, J., Mager, J., Chen, Y., Schneider, E., Cross, J.C., Nagy, A., and Magnuson, T. (2001). Imprinted X inactivation maintained by a mouse Polycomb group gene. *Nature genetics* *28*, 371-375.

Wang, Y., Zhong, Y., Zhou, Y., Tanaseichuk, O., Li, Z., and Zhao, J.C. (2019). Identification of a Xist silencing domain by Tiling CRISPR. *Sci Rep* *9*, 2408.

Wutz, A., and Jaenisch, R. (2000). A shift from reversible to irreversible X inactivation is triggered during ES cell differentiation. *Molecular cell* *5*, 695-705.

Wutz, A., Rasmussen, T.P., and Jaenisch, R. (2002). Chromosomal silencing and localization are mediated by different domains of Xist RNA. *Nature genetics* *30*, 167-174.

Zylicz, J.J., Bousard, A., Zumer, K., Dossin, F., Mohammad, E., da Rocha, S.T., Schwalb, B., Syx, L., Dingli, F., Loew, D., *et al.* (2019). The Implication of Early Chromatin Changes in X Chromosome Inactivation. *Cell* *176*, 182-197 e123.

2nd Editorial Decision

19 July 2019

Thank you for the submission of your revised manuscript to our editorial offices. We have now received the reports from the three referees that were asked to re-evaluate your study, you will find below. As you will see, referees #1 and #2 now support the publication of your manuscript in EMBO reports. Referee #3 has a remaining concern we ask you to address or discuss in a final revised version of your manuscript. Please also provide a detailed response letter addressing these concerns (p-b-p-response). Please also incorporate the text suggestions of referee #1.

Further, I have these editorial requests:

- Please add page numbers to the Appendix, and list the single items in the TOC with the respective page number.
- Please add a paragraph to the Methods section detailing the statistics used throughout the manuscript.
- Please remove the access details for the referees from the data availability section. The information here should be the final one, so that readers can access the published files.
- Please revise the synopsis image (attached in its final size how it will appear online). Please enlarge this, simplify the details, and use bigger fonts. This needs to be in jpeg or tiff format with the exact width of 550 pixels and a height of not more than 400 pixels.
- There are columns in the bar diagrams in Fig. 1C, 4A and EV2B that have no statistics (p-values). Could that be added? Or, if there the differences are not significant, could this be indicated?
- Please add the legends for the dataset and the EV tables as a new TAB to the respective excel files (as first TAB), and then remove these legends from the manuscript text file.
- It seems that the image in Figure 4B (XistFL) is part of the same panel in Fig EV1C. Please indicate this in the legend.
- There is no legend for Fig. EV5H. Please add this.
- We now strongly encourage the publication of original source data (in particular of Western blots) with the aim of making primary data more accessible and transparent to the reader. The source data will be published in a separate source data file online along with the accepted manuscript and will be linked to the relevant figure. If you would like to use this opportunity, please submit the source data (for example scans of entire gels or blots, data points of graphs in an excel sheet, additional images, etc.) of your key experiments together with the revised manuscript. Please include size markers for scans of entire gels, label the scans with figure and panel number. Please send one PDF file per figure.
- We now mandate that all corresponding authors list an ORCID digital identifier that is linked to their EMBO reports account. Please do that for the co-corresponding author (Edith Heard).
- Finally, please find attached a word file of the manuscript text (provided by our publisher) with changes we ask you to include in your final manuscript text, and some queries, we ask you to address. Please provide your final manuscript file with track changes, in order that we can see the modifications done.

REFEREE REPORTS

Referee #1 (referee #2 of previous submission):

I feel that the authors have made sufficient revisions or rebuttals for acceptance of the manuscript.

They added:

Recently, a PcG-defective Xist Δ B mutant also show a reduction of transcriptional silencing at day 14 of differentiation, although the defects could be partially explained by inefficient Xist spreading

along the X chromosome and are due to maintenance function of PcG rather than initiation (Colognori et al., 2019). In any case, in all these systems, it is clear that PcG recruitment will result in some degree of relaxation of transcriptional silencing which...

I believe the tense of show should be showed?
More importantly, "it is clear that PCG recruitment will result ... in relaxation of silencing" should that be abrogation of PCG recruitment??

Referee #2 (referee #3 of previous submission):

I carefully read the replies of the authors to the points that I raised in my original review of this interesting manuscript.

All the points were dealt with in a satisfactory manner and I have no further suggestions to the authors.

Referee #3 (referee #1 of previous submission):

I appreciate the authors diligence in addressing the concerns raised in the prior set of reviews. The issue, however, remains that there are residual H2AK119Ub and H3K27me3 marks on the inactive-X in the Xist $\Delta B+C$ mutant cells. The presence of the low amount of these marks cannot be excluded as the reason that there isn't a significant defect in X-linked gene silencing in the Xist $\Delta B+C$ mutant cells. The authors suggest that the residual H2AK119Ub and H3K27me3 marks are pre-existing marks prior to Xist RNA induction. And, that the loss of new H2AK119Ub and H3K27me3 marks accounts for the defect in gene silencing on the inactive-X, which, the authors themselves note, is rather subtle.

2nd Revision - authors' response

29 July 2019

Referee #1 (referee #2 of previous submission):

I feel that the authors have made sufficient revisions or rebuttals for acceptance of the manuscript.

We thank the referee for all the suggestions and for the positive evaluation of our revised manuscript.

They added:

Recently, a PcG-defective Xist ΔB mutant also show a reduction of transcriptional silencing at day 14 of differentiation, although the defects could be partially explained by inefficient Xist spreading along the X chromosome and are due to maintenance function of PcG rather than initiation (Colognori et al., 2019). In any case, in all these systems, it is clear that PcG recruitment will result in some degree of relaxation of transcriptional silencing which...

I believe the tense of show should be showed?
More importantly, "it is clear that PCG recruitment will result ... in relaxation of silencing" should that be abrogation of PCG recruitment??

This has now been corrected in the manuscript exactly as suggested by the referee.

Referee #2 (referee #3 of previous submission):

I carefully read the replies of the authors to the points that I raised in my original review of this interesting manuscript.

All the points were dealt with in a satisfactory manner and I have no further suggestions to the authors.

We thank the referee for all the suggestions and for the positive evaluation of our revised manuscript.

Referee #3 (referee #1 of previous submission):

I appreciate the authors diligence in addressing the concerns raised in the prior set of reviews. The issue, however, remains that there are residual H2AK119Ub and H3K27me3 marks on the inactive-X in the *Xist* $\Delta B+C$ mutant cells. The presence of the low amount of these marks cannot be excluded as the reason that there isn't a significant defect in X-linked gene silencing in the *Xist* $\Delta B+C$ mutant cells. The authors suggest that the residual H2AK119Ub and H3K27me3 marks are pre-existing marks prior to *Xist* RNA induction. And, that the loss of new H2AK119Ub and H3K27me3 marks accounts for the defect in gene silencing on the inactive-X, which, the authors themselves note, is rather subtle.

We thank the referee for appreciating our efforts at improving the first version of the manuscript. The referee pointed out that residual H2AK119ub and H3K27me3 marks remains in the inactive X-chromosome and this could not be excluded as the reason why no major defects in X-linked gene silencing are observed in the *Xist* $\Delta B+C$ mutant expressing cells. The residual Polycomb (PcG) marks we document in this manuscript are not pre-existing marks on the X chromosome prior to *Xist* RNA induction as the referee mentioned. They are rather observed around X-linked genes upon induction of the PCGF3/PCGF5-PRC1 unbound *Xist* $\Delta B+C$ RNA (Fig. 3B-C). Indeed, such residual Polycomb (PcG) marks on X-linked genes could, in principle, be sufficient to trigger gene silencing. Our data on Fig.4F distinguishes between X-linked genes with or without accumulation of residual PcG marks on their promoters. For the genes with no accumulation, it is fair to say that silencing has been initiated without the need of PcG marks; the same might not hold for the X-linked genes with residual PcG accumulation. Our favorite hypothesis is that PcG recruitment at these genes might be caused through a passive mechanism in response to the initiation of gene silencing by other factors brought by *Xist* (e.g. SPEN) (Fig. 5), since *Xist* $\Delta B+C$ RNA seems not to be able to bind PcG proteins based on our ChIRP-MS data (Fig.2). Indeed, passive recruitment of PcG marks upon silencing has been suggested for autosomal regions in several contexts (Davidovich et al., 2013; Mendenhall et al., 2010; Riising et al., 2014). However, we agree with the referee that we cannot underestimate the possibility that residual accumulation of PcG marks in some X-linked genes might play a role in their silencing. Taking this consideration from the referee, we decide to acknowledge this possibility by adding a sentence (marked in red) in the Discussion section in the following context: *Taking advantage of comparable nChIP-seq and RNA-seq data sets in *Xist* FL and *Xist* $\Delta B+C$, we detected multiple genes which have been silenced and yet with negligible accumulation of H3K27m3 or H2AK119ub at their promoter regions in *Xist* $\Delta B+C$ induced cells (Fig. 4F). This suggests that *Xist*-mediated gene silencing can occur in the absence of PcG recruitment, at least, for a subset of X-linked genes. The other subset of X-linked genes have detectable residual accumulation of PcG marks, which could play a role in the initiation of their silencing. Alternatively, the recruitment of PcG at X-linked genes could be secondary to transcriptional silencing in PCGF3/5-PRC1-unbound *Xist* $\Delta B+C$. It has been proposed that PcG recruitment to active promoters and gene bodies could be passive upon their silencing, in the sense that the PcG system will operate on any transcriptionally inactive, GC-rich locus (Davidovich et al., 2013; Mendenhall et al., 2010; Riising et al., 2014) (Fig. 5).*

References:

Davidovich, C., Zheng, L., Goodrich, K.J., and Cech, T.R. (2013). Promiscuous RNA binding by Polycomb repressive complex 2. *Nat Struct Mol Biol* 20, 1250-1257.

Mendenhall, E.M., Koche, R.P., Truong, T., Zhou, V.W., Issac, B., Chi, A.S., Ku, M., and Bernstein, B.E. (2010). GC-rich sequence elements recruit PRC2 in mammalian ES cells. PLoS Genet 6, e1001244.

Riising, E.M., Comet, I., Leblanc, B., Wu, X., Johansen, J.V., and Helin, K. (2014). Gene silencing triggers polycomb repressive complex 2 recruitment to CpG islands genome wide. Molecular cell 55, 347-360.

Corresponding Author Name: Simão José Teixeira da Rocha

Manuscript Number: EMBOR-2019-48019